# Engineered reporter phages for detection of *Escherichia coli, Enterococcus,* and *Klebsiella* in urine

Susanne Meile [1,6], Jiemin Du [1,6], Samuel Staubli [1], Sebastian Grossmann [1], Hendrik Koliwer-Brandl [2], Pietro Piffaretti[1], Lorenz Leitner[3], Cassandra I. Matter [1], Jasmin Baggenstos[1], Laura Hunold[1], Sonja Milek[3], Christian Guebeli [1], Marko Kozomara-Hocke[4], Vera Neumeier[3], Angela Botteon[1], Jochen Klumpp[1], Jonas Marschall[5], Shawna McCallin[3], Reinhard Zbinden[2], Thomas M. Kessler [3], Martin J. Loessner [1], Matthew Dunne [1] ✉ & Samuel Kilcher [1] ✉

The rapid detection and species-level differentiation of bacterial pathogens facilitates antibiotic stewardship and improves disease management. Here, we develop a rapid bacteriophage-based diagnostic assay to detect the most prevalent pathogens causing urinary tract infections: *Escherichia coli, Enterococcus* spp., and *Klebsiella* spp. For each uropathogen, two virulent phages were genetically engineered to express a nanoluciferase reporter gene upon host infection. Using 206 patient urine samples, reporter phage-induced bioluminescence was quantified to identify bacteriuria and the assay was benchmarked against conventional urinalysis. Overall, *E. coli, Enterococcus* spp., and *Klebsiella* spp. were each detected with high sensitivity (68%, 78%, 87%), specificity (99%, 99%, 99%), and accuracy (90%, 94%, 98%) at a resolution of ≥10$^3$ CFU/ml within 5 h. We further demonstrate how bioluminescence in urine can be used to predict phage antibacterial activity, demonstrating the future potential of reporter phages as companion diagnostics that guide patient-phage matching prior to therapeutic phage application.

Urinary tract infections (UTIs) are among the most common community-acquired and healthcare-associated bacterial infections in all age groups and represent a major public health problem worldwide[1]. The societal and economic burden of UTIs is substantial, as evidenced by its global prevalence of around 150 million cases per year and an estimated direct healthcare expenditure of over 6 billion dollars[1]. Furthermore, UTIs aggravate the antimicrobial resistance crisis as they constitute a leading cause of antibiotic prescription, second only to respiratory tract infections[2]. While uncomplicated UTIs generally present minimal morbidity, the presence of certain predisposing medical conditions (e.g., metabolic disorders, immunosuppression, and obstructive uropathy) can lead to the development of life-threatening complications such as pyelonephritis and urosepsis[3,4]. The etiology of UTI is complex, often polymicrobial, and can involve a plethora of bacterial species and certain fungi. Gram-negative bacteria, predominantly from the *Enterobacteriaceae* family, including *Escherichia coli* and *Klebsiella* spp., cause the majority of UTI cases, closely followed by *Enterococcus faecalis* as the major Gram-positive uropathogen[5,6].

[1]Institute of Food Nutrition and Health, ETH Zurich, Zurich, Switzerland. [2]Institute of Medical Microbiology, University of Zurich, Zurich, Switzerland. [3]Department of Neuro-Urology, Balgrist University Hospital, University of Zurich, Zurich, Switzerland. [4]Department of Urology, University Hospital Zurich, Zurich, Switzerland. [5]Division of Infectious Diseases, Department of Medicine, Washington University School of Medicine, St. Louis, MO, US. [6]These authors contributed equally: Susanne Meile, Jiemin Du. ✉e-mail: matthew.dunne@hest.ethz.ch; samuel.kilcher@hest.ethz.ch

Due to this diverse etiology, rapid and accurate pathogen identification is of paramount importance to guarantee a timely and effective therapy and to limit the preventive prescription of broad-range antibiotics.

The gold standard for diagnosing UTIs is the presence of symptoms combined with a positive urine culture[7]. Diagnostic accuracy can be further improved by integrating mass spectrometry identification and/or 16s rRNA sequencing, when required[7]. Despite providing quantitative results with species-level resolution, culture-based assays require a minimum of 18–30 h to return clinically actionable results[7]. Long turnaround times propel the empirical prescription of antibiotics, which, in turn, aggravates the global spread of antimicrobial resistance. To tackle this issue, multiple immunological and molecular diagnostic techniques have been developed to enable uropathogen detection in point-of-care (POC) settings[7]. However, they either present poor specificity and/or sensitivity (e.g., urine dipsticks), require considerable laboratory resources (e.g., flow cytometry), or fail to discriminate between dead and viable cells (e.g., multiplex PCR)[7]. Most of these methods cannot identify the causative agent and some are not suitable for Gram-positive pathogen detection (e.g., dipstick nitrite test). Therefore, an alternative POC diagnostic, which combines the desired 4 S properties (simplicity, speed, sensitivity, and specificity), could provide substantial value to UTI management beyond what is currently available.

Bacteriophages (phages) are viral predators of bacteria that bind and infect their host cells with genus-, species-, or strain-level specificity. The clinical use of phages as pathogen-specific antimicrobials (phage therapy) is a promising strategy to tackle multi drug-resistant (MDR) infections[8] and to manage microbial dysbiosis[9]. By leveraging their specific interaction with bacterial cells, phages have also been used to develop an array of diagnostic tools for rapid and accurate pathogen identification (reviewed elsewhere[10]). For example, phages can be genetically engineered to encode a heterologous reporter gene —a concept known as "reporter phage". Upon infection of the host bacterium, phage-driven reporter gene expression is detected through enzymatic substrate conversion, indicating the presence of viable host cells. Nanoluciferase (NLuc)—an engineered luciferase derived from deep-sea shrimp—is one of the most promising genes for reporter phage engineering. NLuc is a small and highly stable enzyme that produces strong, ATP-independent, glow-type luminescence upon substrate addition[11]. NLuc reporter phages can reliably detect bacterial contaminations at very low concentrations[12–14].

In this study, we isolate suitable phage backbones, develop the required engineering technology to construct reporter phages, and devise a simple and accurate NLuc reporter phage-based POC diagnostic tool. Compared to the gold standard, this assay demonstrates rapid and specific detection of the three most prevalent uropathogens from 206 fresh clinical urine specimens. Besides supporting clinical decision-making, we use patient isolates to demonstrate the potential application of reporter phages as companion diagnostics to identify responder patients that could benefit from personalized therapeutic phage application.

## Results

### Isolation and characterization of phages against *E. coli*, *Klebsiella*, and *Enterococcus*

To validate UTI etiology, we collected 663 uropathogen isolates from 442 incidents of either UTI (227 incidents) or asymptomatic bacteriuria (215 incidents) in Zurich, Switzerland (the Zurich Uropathogen Collection Figs. 1a, S1, Table S1). We identified 30 different bacterial species, with *E. coli* (31%), *Enterococcus* spp. (20%), and *Klebsiella* spp. (18%) as the three most prevalent uropathogens, which is consistent with previous etiological studies on UTIs[5]. Strictly lytic phages against each of the top three uropathogens were isolated from different wastewater sources across Switzerland (Fig. 1a) by sequential phage propagation on multiple strains of the respective host species, which

provided selective pressure towards isolating phages with a broad host range[15]. To ensure activity on uropathogenic strains, sequential isolation was performed using UTI isolates from the Zurich Uropathogen Collection. Prior to performing plaque assays, a short enrichment step in synthetic human urine medium (SHU, pH 6) was included for each host to select for phage activity in high urea-containing environments[16].

By applying this stringent selection process, we isolated phages E2 and E4 (targeting *E. coli*), K1 and K4 (targeting *Klebsiella* spp.), and EfS3 and EfS7 (targeting *Enterococcus* spp.). All phages were de novo sequenced, visualized by electron microscopy, and their plaquing host ranges were determined on more than 50 urological isolates. A summary of phage characteristics is presented in Fig. 1b. E2, E4, K1, and K4 are members of the *Tevenvirinae* subfamily within the *Straboviridae* family. Accordingly, these phages feature circularly permuted dsDNA genomes of about 170 kbp that are injected through a characteristic contractile tail. Phage EfS3 is a member of the *Herelleviridae* family with a contractile tail and a genome size of 152,305 bp, while EfS7 has an elongated head and a non-contractile tail and belongs to the *Saphexavirus* genus with a 56,144 bp unit genome. The combined plaquing host range for each phage pair was broad, covering 71% (*E. coli*), 60% (*K. pneumoniae* and *K. oxytoca*), and 77% (*E. faecalis*) of the tested urological isolates (see in vitro assessment).

### Engineering of nanoluciferase reporter phages

Due to their broad host ranges, these six phages were used as scaffolds to engineer NLuc reporter phages for *E. coli*, *Klebsiella* spp., and *Enterococcus* spp. detection. The *nluc* sequence was integrated alongside a ribosomal binding site (RBS) downstream of phage structural genes (Fig. 2a). Because the structural gene cassette is strongly expressed from endogenous phage promoters, this strategy confers a high level of reporter gene expression. Homologous recombination-based and CRISPR-Cas9-assisted phage engineering has previously been achieved for *Tevenvirinae*[17–19] and was adopted for the construction of phages E2::*nluc*, E4::*nluc*, K1::*nluc*, and K4::*nluc* (see methods section for more detail). In contrast, no established technology was available for genetic engineering of virulent *Enterococcus* phages, and we therefore developed a two-step protocol as shown in Fig. 2b–d (workflow shown in Fig. S2a).

In the first step, *Enterococcus* phages were propagated in the presence of an editing template vector (pEdit) containing the *nluc* gene and flanking regions of homology that direct sequence-specific integration (pEDIT_EfS3 and pEDIT_EfS7, respectively). Resulting phage lysates contain a mixed population with few recombinants and a majority of wildtype phages. In the second step, these lysates were subjected to counterselection on a host encoding an episomal CRISPR-Cas system (pSelect) designed to cleave wildtype phage genomes without targeting recombinants, which enables selective enrichment of recombinant phages. The plasmids pSelect_EfS3 and pSelect_EfS7 contain the *S. pyogenes cas9* gene, tracrRNA, and a crRNA element with two spacers (S1 and S2) (Fig. 2b), which target the two regions flanking the *nluc* insertion site of EfS3 and EfS7, respectively. Recombinant genomes are protected from counterselection through the integration of silent mutations within the protospacer adjacent motifs (PAMs) on pEdit (see Fig. 2c). This CRISPR-Cas9 system completely restricted wildtype EfS3 and EfS7 plaque formation on *E. faecalis* (Figs. 2b and S2b, c) and enabled rapid identification of recombinant, CRISPR-resistant phage candidates (Fig. 2c, d), all of which featured the expected genotype and bioluminescence (Fig. 2d). EfS3::*nluc* and EfS7::*nluc* were plaque purified alongside the *E. coli* and *Klebsiella* reporter phages and sequenced to confirm reporter gene integration.

### In vitro functional assessment of reporter phages

To guide the development of our reporter phage urine assay, we used in vitro experiments to determine key functional characteristics of all

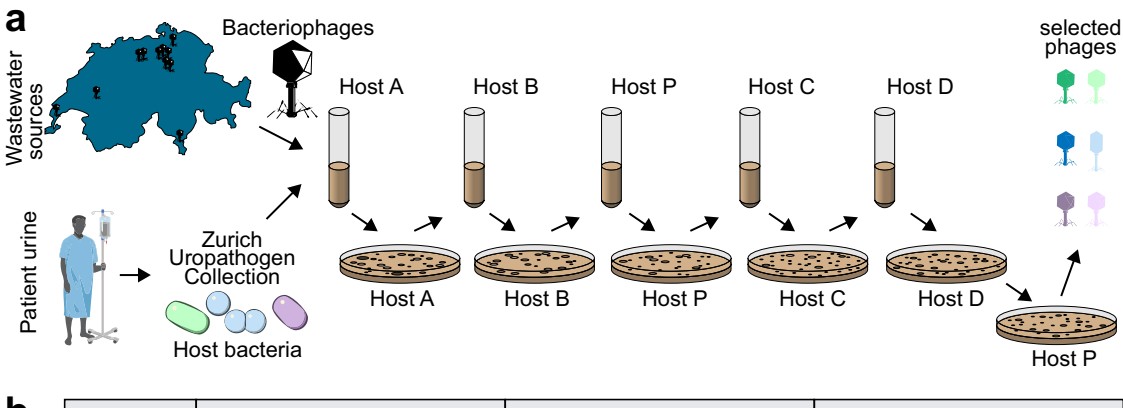

| Phage | E2 | E4 | EfS3 | EfS7 | K1 | K4 |
|---|---|---|---|---|---|---|
| Genus | *Tequatrovirus* | | *Shiekvirus* | *Saphexavirus* | *Jiaodavirus* | |
| Genome size | 166.4 kb | 169.2 kb | 152.3 kb | 56.1 kb | 170 kb | 168.8 kb |
| GenBank No. | OL870316 | OL870317 | OL870611 | OL870612 | OL870318 | OL870319 |
| Bacterial target | *E. coli* | | *Enterococcus* spp. | | *Klebsiella* spp. | |
| Morphology | | | | | | |

**Fig. 1 | Isolation and characterization of strictly lytic *E. coli*, *Enterococcus* spp. and *Klebsiella* spp. phage scaffolds. a** Workflow depicts the sequential multiple host method for the isolation of phages with broad host-range. Wastewater samples from different wastewater treatment plants across Switzerland were mixed with the first clinical isolate (Host A) of the respective target species selected from the Zurich Uropathogen Collection and briefly enriched in synthetic human urine medium (SHU) before performing plaque assays. Emerging plaques were pooled, extracted, and propagated on 4 subsequent hosts (B, P, C, D) before individual phages were purified on a propagation Host P. **b** Summary of the genomic and morphological characterization of six selected phage scaffolds infecting *E. coli* (phages E2 and E4), *Enterococcus* spp. (phages EfS3 and EfS7) and *Klebsiella* spp. (phages K1 and K4), respectively. Some elements of the figure were created using biorender.com. Source data are provided as a Source Data file.

engineered phages, including infection kinetics, detection limits, detection range, and potential cross-reactivities. All phage lysates were purified by polyethylene-glycol precipitation and caesium chloride (CsCl) gradient ultracentrifugation to remove residual NLuc enzyme present within raw phage lysates.

Initial phenotypic characterization of our reporter phages revealed no significant alterations in plaque morphology (Fig. 3a) or host-cell lysis (Fig. 3b) when compared to their parental wildtype counterparts, suggesting that *nluc* gene insertion is not associated with major fitness costs. To determine the time required to reach a stable endpoint bioluminescence signal, we performed kinetic infection experiments. To this end, individual reporter phages were monitored during infection of their production hosts by measuring bioluminescence in relative light units (RLU) over 4–5 h. As shown in Fig. 3c, all reporter phages presented a steep rise in luminescence intensity almost immediately after the onset of infection with a stable plateau of RLU fold change (FC) above $10^5$ reached between 1.5 and 3 h post infection (p.i.) for *E. coli* and *Klebsiella* and a plateau with RLU FC $10^4$ between 2 and 4 h for *Enterococcus*. While these kinetic profiles are promising, they can exhibit variations depending on the infected strain and on the sample matrix. For example, urine composition and pH can vary significantly between patients[20]. Hence, to ensure measurement of stably saturated signals in patient urine, all bioluminescence assays were measured at 3 h p.i. for *E. coli* and *Klebsiella* or at 4 h p.i. for *Enterococcus*. To determine the detection limit of our assay, reporter phage infections were performed on serial dilutions of host bacteria. The detection limit was defined as the minimum number of colony-forming units (CFUs) required to produce a signal above the background threshold (i.e., average background RLUs plus three-fold standard deviations (SD)). Without any pre-enrichment of bacteria,

detection limits below 100 CFU/ml were observed for all reporter phages (Fig. 3d).

The bioluminescence detection range of our reporter phages was tested against a panel of 154 randomly selected urological isolates (52 *E. coli*, 52 *E. faecalis*, and 50 *Klebsiella* spp.) and compared to phage plaquing ability, as determined by spot-on-the-lawn assays (Fig. 4a). While bioluminescence demonstrates the ability of a phage to inject genetic material into the bacterial cytoplasm and express luciferase, the criterion for plaquing is more stringent, as the phage must complete a full infection cycle and lyse the bacterial cell[21]. Each individual phage exhibited a broad bioluminescence detection range of 81% (E2::*nluc* and E4::*nluc*), 71% (EfS3::*nluc*), 48% (EfS7::*nluc*), and 84% (K1::*nluc*), 80% (K4::*nluc*). Despite significant overlaps, complementation of the detection range was observed for all phage pairs. Accordingly, the combined range of detection for *E. coli*, *E. faecalis*, and *Klebsiella* spp. could be improved to 83%, 96%, and 90%, respectively. All phages presented significantly broader bioluminescence detection ranges compared to their plaquing ability (Fig. 4b), which could be due to bacterial intracellular defense systems that prevent the completion of a full infection cycle in some strains[22].

Low diversity within strain collections can lead to an over- or underestimation of the phage host range. We addressed the clonal diversity of our host range panel (i) by analyzing the signal intensity distributions in response to reporter phage infection, and (ii) by comparing antibiotic resistance profiles within the *E. coli* strain panel. Luminescence intensities were distributed over several orders of magnitude (Fig. S3f), suggesting that the selected strains differ significantly in phage-susceptibility. Analysis of antibiotic susceptibility patterns from *E. coli* revealed the presence of at

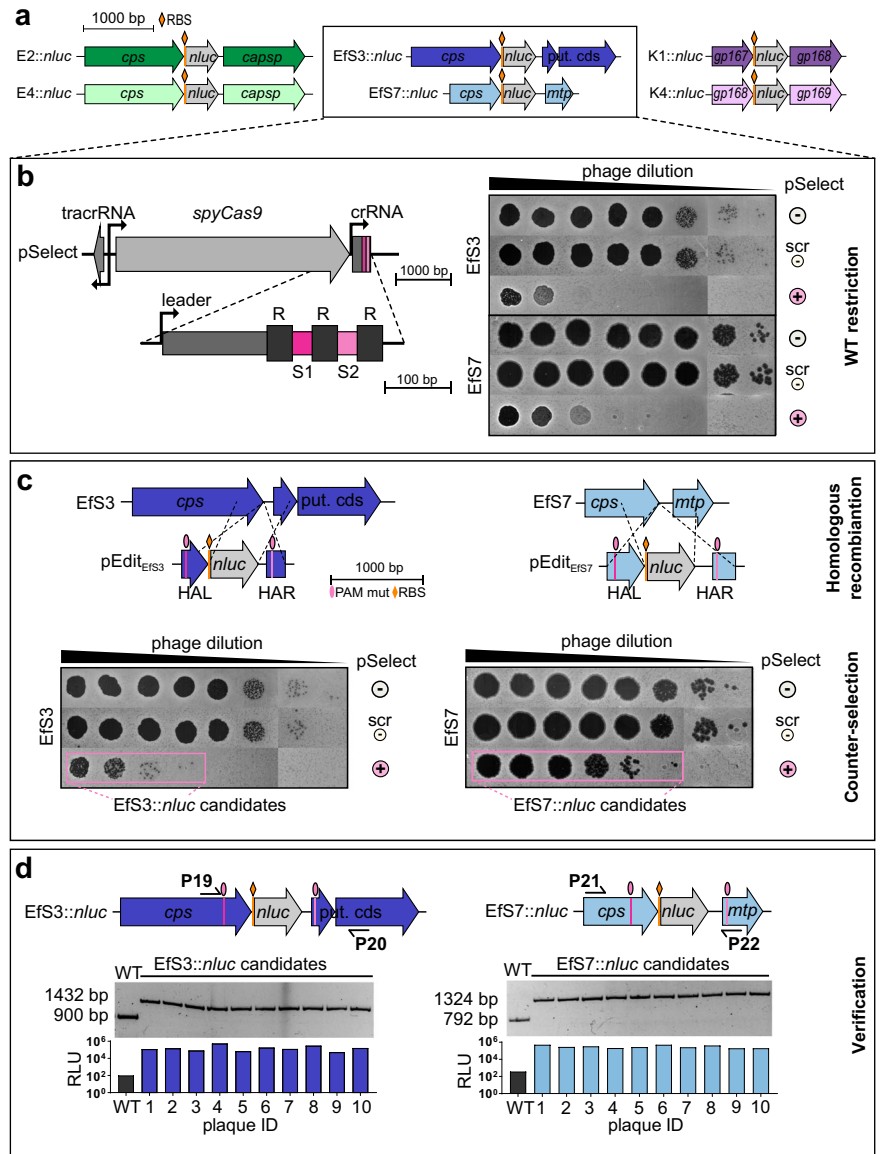

**Fig. 2 | Engineering of virulent *E. coli*, *Enterococcus* and *Klebsiella* reporter phages. a** Schematic representation of the nanoluciferase gene (*nluc*) insertion sites in the genomes of all reporter phages. For *E. coli* and *Enterococcus* phages, codon optimized *nluc* (516 bp + RBS) was incorporated immediately downstream of the major capsid gene (*cps*), whereas for *Klebsiella* phages, *nluc* was inserted downstream of the prohead assembly protein (gp167 for K1 and gp168 for K4). **b**–**d** Engineering method for virulent *Enterococcus* phages combining homologous recombination with CRISPR-Cas9-assisted counterselection. **b** Left: Schematic representation of the pSelect CRISPR locus used to restrict wildtype phage genomes by targeting two sequences flanking the *nluc* integration site. Right: Spot-on-the-lawn assays of 10-fold phage dilutions show plaquing efficiency on hosts with (+) or without (scr, -) CRISPR targeting. scr = non-targeting crRNA; - = no pSelect.

**c** Schematic representation of the editing plasmids (pEdit_EfS3, pEdit_EfS7) used to incorporate mutated protospacer-adjacent motifs (PAMs) and *nluc* gene sequences downstream of the major capsid protein (*cps*) regions of EfS3 and EfS7 by homologous recombination and subsequent CRISPR escape of EfSs::*nluc* and EfS7::*nluc* candidates (counterselection) shown by spot-on-the-lawn assays of serial dilutions of phages after recombination. **d** 10 individual plaques were isolated from one replicate of data shown in Fig. S2b, c. Plaques were validated genetically and functionally using PCRs with primers P19/20 and P21/22 and RLU determination from individual plaques, respectively. HAL = homology arm left; HAR = homology arm right; put. cds = putative coding sequence; mtp = major tail protein; RLU = relative light unit. Source data are provided as a Source Data file.

least 24 different resistance profiles in 52 strains (Fig. S3g). Taken together, these data suggest a high level of clonal diversity across tested isolates.

Since bacterial UTIs are of diverse etiology and frequently poly-microbial in nature, it is important to ensure that the reporter phages exhibit high specificity toward their target bacterial species. While false-positives are best assessed in a large patient study (as shown in Fig. 5), an initial cross-reactivity bioluminescence assay was performed in vitro using 67 non-target bacterial isolates, including representative strains from different uropathogenic species. No cross-reactivity was observed for E2::*nluc* and E4::*nluc*; however, K1::*nluc* and K4::*nluc*

did cross-react with one clinical isolate of *Raoultella ornithinolytica* (Fig. S3d). The genus *Raoultella* was formerly a cluster within the genus *Klebsiella*, but was recently reclassified based upon advanced phylo-genetic analysis[23]. Nevertheless, the cross-reactivity observed can only be explained by the *Raoultella* isolate presenting functional cell surface receptors for phages K1 and K4. The *Enterococcus* reporter phages EfS3::*nluc* and EfS7::*nluc* revealed genus-specific infectivity, however, EfS3::*nluc* produced bioluminescence (but no plaques) on 2/12 tested *S. aureus* strains and a single *Pediococcus acidilactici* isolate (Fig. S3b, c). In addition, we demonstrated that phage EfS3::*nluc* is able to infect 9 out of 14 tested *E. faecium* strains, whereas both the plaquing- and detection

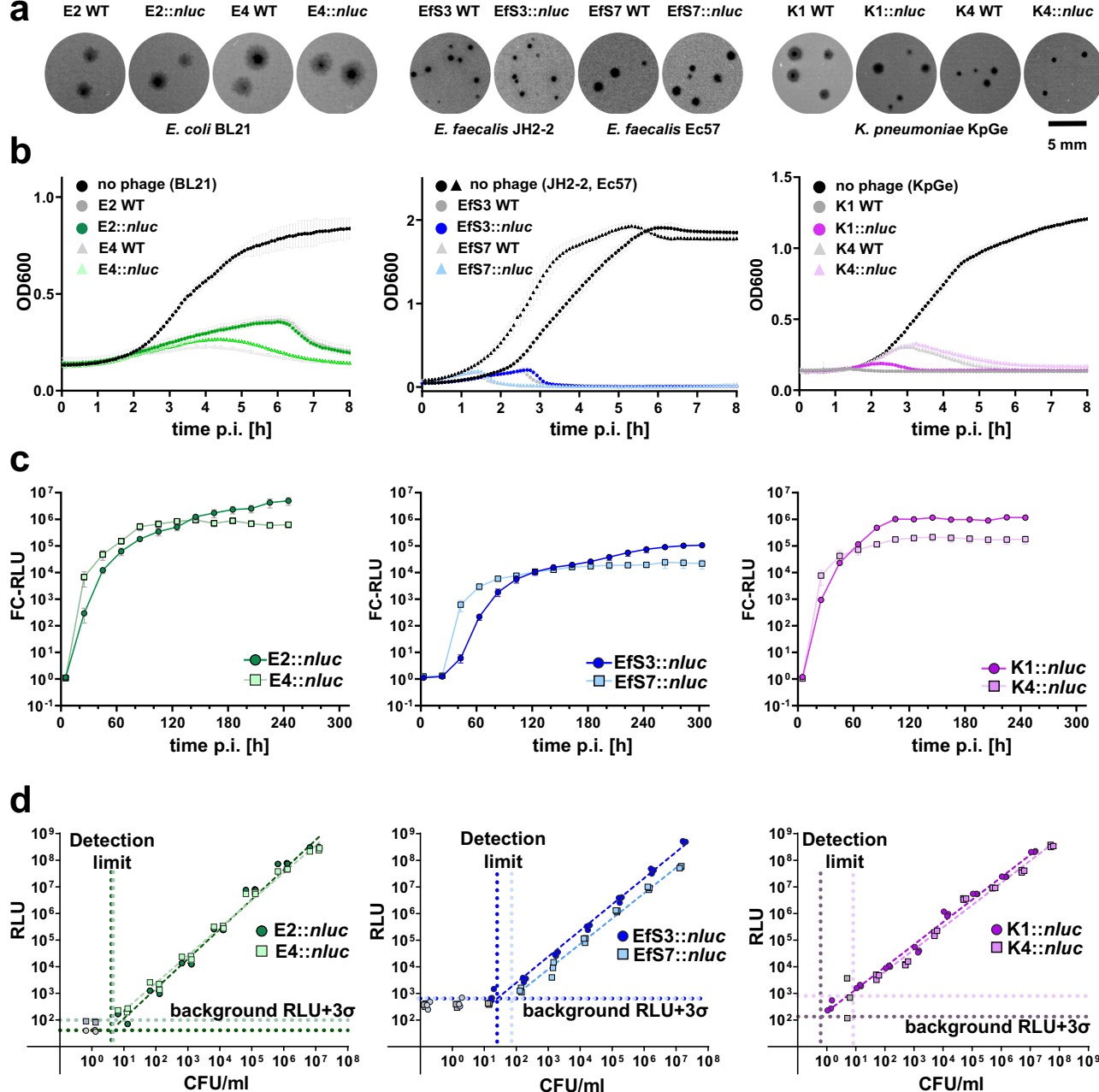

**Fig. 3 | In vitro functional assessment of the reporter phages E2::*nluc*, E4::*nluc*, EfS3::*nluc*, EfS7::*nluc*, K1::*nluc* and K4::*nluc*. a** Plaque phenotypes of wildtype and recombinant reporter phages were compared using the soft-agar overlay method. **b** Infection kinetics of engineered vs. wildtype phages were compared by quantifying the optical density ($OD_{600}$) of infected bacterial cultures over time (liquid infection assay). Data are mean ± SD ($n = 3$). **c** The kinetics of reporter phage-mediated NLuc signal production were determined using bioluminescence time course assays. Fold-change (FC) RLU equals RLU (sample) divided by RLU (phage only control). Data and mean±standard error of the mean (SEM) from biological triplicates. **d** Assay sensitivity was determined in vitro for each reporter phage by infecting serial host cell dilutions and quantifying luminescence at 3 h (*E. coli* and *Klebsiella* phages) or 4 h (*Enterococcus* phages) post infection. Detection limits (vertical dotted lines) were calculated as the cell number required to produce a signal that is three standard deviations (3σ) above the background luminescence (phage in medium, indicated as horizontal dotted lines). Datapoints for linear regression are from biological triplicates. All infections were performed on the phage propagation hosts. p.i. post infection; RLU relative light unit. Source data are provided as a Source Data file.

range of phage EfS7::*nluc* did not include *E. faecium* (Fig. S3a). Generally, species-level specificity is less critical for enterococci because *E. faecium* is very rarely detected in patient urine[24].

To assess infection and reporter protein expression under relevant conditions, we tested each reporter phage in pooled human urine spiked with six different clinical isolates. Infectivity and bioluminescence generation was maintained in 34/36 patient isolate-phage combinations, suggesting urine-dependent matrix effects do not

constitute a major hurdle for assay development (Fig. 4c). Besides UTIs, enterococci are a frequent cause of Gram-positive bacteraemia and sepsis[25]. Therefore, we tested our *Enterococcus* reporter phages in whole human blood spiked with *E. faecalis* strain Ef24 and compared signal intensity and kinetics against a comparable experiment with spiked human urine (Fig. 4d, e). Although bioluminescence signal generation was slightly delayed compared to urine, both EfS3::*nluc* and EfS7::*nluc* enabled robust detection of $10^4$ CFU/ml within 4 h in whole

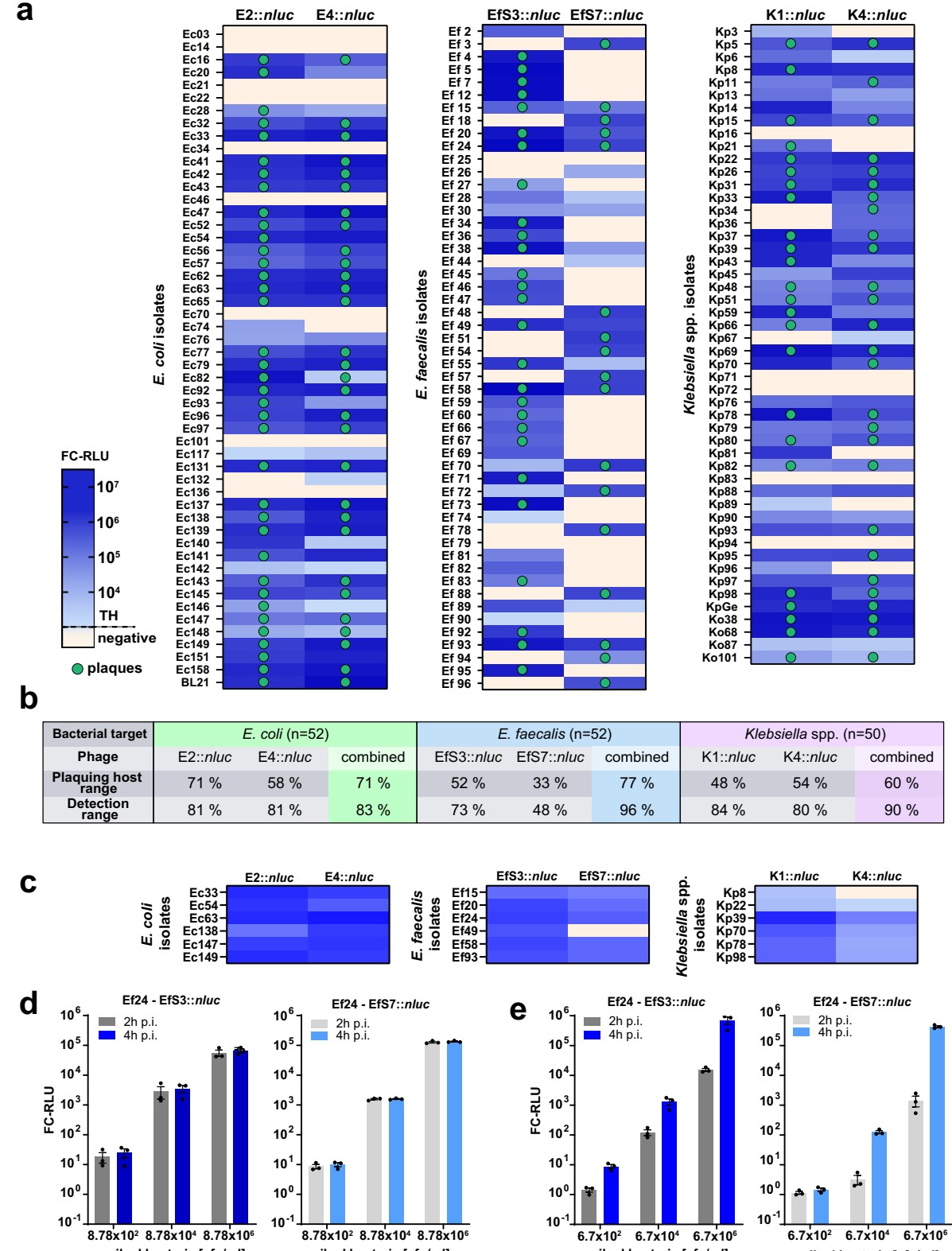

**b**

| Bacterial target | *E. coli* (n=52) | | | *E. faecalis* (n=52) | | | *Klebsiella* spp. (n=50) | | |
|---|---|---|---|---|---|---|---|---|---|
| Phage | E2::*nluc* | E4::*nluc* | combined | EfS3::*nluc* | EfS7::*nluc* | combined | K1::*nluc* | K4::*nluc* | combined |
| Plaquing host range | 71 % | 58 % | 71 % | 52 % | 33 % | 77 % | 48 % | 54 % | 60 % |
| Detection range | 81 % | 81 % | 83 % | 73 % | 48 % | 96 % | 84 % | 80 % | 90 % |

blood (Fig. 4e). *Enterococcus* detection in blood was confirmed with an additional set of five clinical isolates (Fig. S3e).

### Reporter phages enable rapid and sensitive detection of *E. coli*, *Klebsiella*, and *Enterococcus* in patient urine

Culture-based detection of $10^5$ CFU/ml in urine is the gold standard (GS) for diagnosis of asymptomatic bacteriuria and UTI, although lower thresholds ($10^3$–$10^4$ CFU/ml) have been suggested, particularly for catheter-associated UTIs (CAUTIs)[24,26]. Based on our in vitro reporter phage characterization, we developed a protocol for rapid and direct detection of *E. coli*, *Klebsiella* spp., and *Enterococcus* spp. in patient urine and tested its performance in a large-scale patient evaluation (workflow: Fig. 5a). To this end, 206 fresh urine samples were collected from two university hospitals in Switzerland (site 1 and site 2)

**Fig. 4 | Reporter phage host range analysis and infectivity in human urine and whole blood. a** Plaquing host ranges and bioluminescence detection ranges were determined in growth medium on 52 urological *E. coli* and *E. faecalis* isolates and 50 urological *Klebsiella* spp. isolates from the Zurich Uropathogen Collection (listed in Table S2). Strains were infected with reporter phages and luminescence quantified at 3 h (*E. coli* and *Klebsiella* spp.) or 4 h (*Enterococcus* spp.) post infection. Heat-map shows fold change RLU (FC-RLU). Threshold (TH): ≥ 100 FC-RLU (*E. coli* and *Klebsiella* spp.) and 50 FC-RLU (*Enterococcus* spp.). Clinical isolates that allow for plaque formation are indicated with a green dot. Plaquing host ranges were determined from single spot on the lawn infection assays and bioluminescence detection ranges were from biological quadruplicates (*E. coli* and *K. pneumoniae*) or biological triplicates (*E. faecalis*). **b** Summary of individual and combined host- and detection ranges. Each reporter phage was able to transduce bioluminescence into strains that were not permissive to plaque formation, effectively leading to an increased detection range compared to the plaquing host range. **c** Heat maps showing reporter phage-induced luminescence in human urine spiked with 6 selected strains of each bacterial target species. **d, e** Dose response assay with *Enterococcus* phages EfS3::*nluc* and EfS7::*nluc* in human urine **d** or blood **e** spiked with serial dilutions of the clinical isolate *E. faecalis* Ef24. After 1/10 dilution of the spiked urine/blood with buffered media and a 1 h enrichment, reactions were mixed with $2 \times 10^7$ PFU/ml reporter phages and FC-RLU was determined at 2 h and 4 h p.i. FC RLU equals RLU (sample) divided by RLU (phage only control). All data in **c**–**e** is shown as mean (±SEM) from biological triplicates. Individual datapoints are shown as black dots. Source data are provided as a Source Data file.

over a period of five months and subjected to reporter phage assays. For benchmarking, all samples were independently analyzed by the Institute of Medical Microbiology (IMM, University of Zurich, Switzerland). In brief, this involved culture-based identification combined with mass spectrometry, where required (i.e., MALDI-ToF-MS)[27,28]. In addition, all samples were analyzed in house by selective and differential plating to complement IMM results.

Prior to phage addition, urine specimens were enriched in growth medium for 1 h to activate the metabolism of stressed bacteria and to reduce possible matrix effects. Individual reporter phages were added to the enriched urine and luminescence was quantified at 3 h p.i (*E. coli*, *Klebsiella* spp.) or 4 h p.i. (*Enterococcus* spp.). A heatmap was generated to compare the reporter phage assay results to selective plating (Fig. 5b). *E. coli* was identified in 59 out of the 206 urine samples through conventional detection (i.e., selective plating), whereas combined results from individual E2::*nluc* and E4::*nluc* assays led to 40 true positive identifications of *E. coli* (68% sensitivity). In contrast, the sensitivity for *E. faecalis* detection by EfS3::*nluc* and EfS7::*nluc* and *Klebsiella* spp. detection by K1::*nluc* and K4::*nluc* was considerably higher at 78% (35/45) and 87% (26/30), respectively. Besides *K. pneumoniae* and *K. oxytoca*, K4::*nluc* also identified one urine sample (specimen 152) with the atypical *Klebsiella* subspecies *K. aerogenes* (previously *Enterobacter aerogenes*)[29]. Interestingly, while all 15 *Klebsiella*-containing samples from Site 1 were detected, the assay failed to identify 4 out of 12 positive samples from Site 2, possibly due to site-specific difference in circulating clones. In total 5 false positives (2.4%) were observed across 206 samples and considering all three target pathogens: Specimens 6, 18, and 37 were likely due to sample cross-contamination; specimen 101 was a result of cross-reactivity of EfS3::*nluc* with *S. xylosus*; sample 146 yielded a false-positive result due to different data evaluation criteria between the reporter phage assay and the GS analysis: Specifically, we detected *Enterococcus* by reporter phage bioluminescence and in-house plating, however, GS analysis returned a negative result because cell numbers remained below the diagnostic threshold of $10^3$ CFU/ml.

In sum, across 206 patient specimens, the reporter phage assay yielded an overall diagnostic sensitivity of 68%, 78%, and 87%, a specificity of 99%, 99%, and 99%, and an accuracy of 90%, 94%, and 98% for *E. coli*, *Enterococcus* spp., and *Klebsiella* spp., respectively (summarized in Fig. 5c). The species distribution of strains identified during the field evaluation was comparable to the Zurich Uropathogen Collection, with *E. coli*, *Klebsiella* spp., and *Enterococcus* spp. within the top five most prevalent uropathogens (Fig. S4).

### Reporter phages as companion diagnostics for phage-patient matching

Therapeutic phage candidates are typically selected based on a broad host range within the bacterial target species. However, individual phages rarely form plaques on more than 70% of patient isolates (see plaquing of phages in Fig. 4a). These differences in antibacterial efficacy are mostly due to the presence of phage defense mechanisms encoded within the host genomes, inevitably resulting in treatment failure[22]. To increase the likelihood of therapeutic success, a rapid strategy is required to identify suitable phages for each individual patient. Such a companion diagnostic assay for phage-patient matching should predict the antibacterial effect against specific patient isolates under relevant conditions (e.g., directly in urine or blood)[30].

Beyond their application for detection and differentiation, the speed and simplicity of reporter phage assays could potentially be harnessed to develop companion diagnostics for phage therapy. However, while reporter phage-induced bioluminescence is a reliable indicator of successful genome delivery, it is unclear whether it can predict killing of patient-derived strains. To address this question, we isolated 50 *E. coli*, 22 *Klebsiella* spp., and 36 *E. faecalis* strains from the field evaluation and quantified bioluminescence in response to reporter phage challenge under defined conditions ($10^7$ CFU/ml, $10^9$ PFU/ml, SHU). These RLU values were compared with results obtained from turbidity reduction assays in SHU as a measure of antibacterial activity (Fig. S5a). To this end, growth curves were monitored over 12 h in the presence or absence of wildtype phage and the relative reduction in area under the curve (AUC ratio = $AUC_{phage}/AUC_{control}$) was quantified. Using these standardized conditions, a RLU vs. AUC ratio dot plot was generated for each pathogen (Fig. S5b), and calculations for precision, sensitivity and F1-score were used to determine RLU and AUC ratio cut-offs that yield the best overall prediction of antibacterial efficacy for individual phage-host pairs (Fig. S5c). The F1-score is the harmonic mean of precision and sensitivity and can be used to optimize trade-offs between these two metrics. Based on these parameter optimizations, true positives were gated at an AUC ratio <0.2 and at RLU values > $10^5$ (*E. coli*), > $10^4$ (*Enterococcus* spp.), or > $10^6$ (*Klebsiella* spp.).

These gating parameters (Fig. S5) were determined in a defined growth medium (SHU) with a defined cell concentration ($10^7$ CFU/ml), which does not reflect a clinical scenario where the patient's urine composition and bacterial cell numbers are unknown. To test how well treatment success can be predicted based only on the raw luminescence values obtained from patient urine, we applied these gating criteria to the luminescence data from the field evaluation (workflow: Fig. 6a). The corresponding dot plots comparing RLU to the AUC ratio are shown for all three pathogens in Fig. 6b–d (right). A selection of activity plots for correctly identified responder patients (true positives) with their matching therapeutic phages are shown in Fig. 6b–d (left). The positive predictive value (i.e., precision) for the companion diagnostic test in patient urine was calculated at 78% (*E. coli*), 93% (*Enterococcus* spp.), and 47% (*Klebsiella* spp.). It can be concluded that RLU is an excellent predictor of antibacterial efficacy for *Enterococcus* and a good predictor for *E. coli*, suggesting that reporter phage data could be employed for therapeutic phage selection. Due to many false-positive predictions, companion diagnostic assay development is more challenging for *Klebsiella* spp. Overall, our companion diagnostic tends to exclude some patients that may benefit from therapy (low sensitivity), which is due to rather stringent selection criteria and/ or low bacterial burden in patient urine resulting in weak signals (=false negatives). Nevertheless, our data suggests that rapid reporter

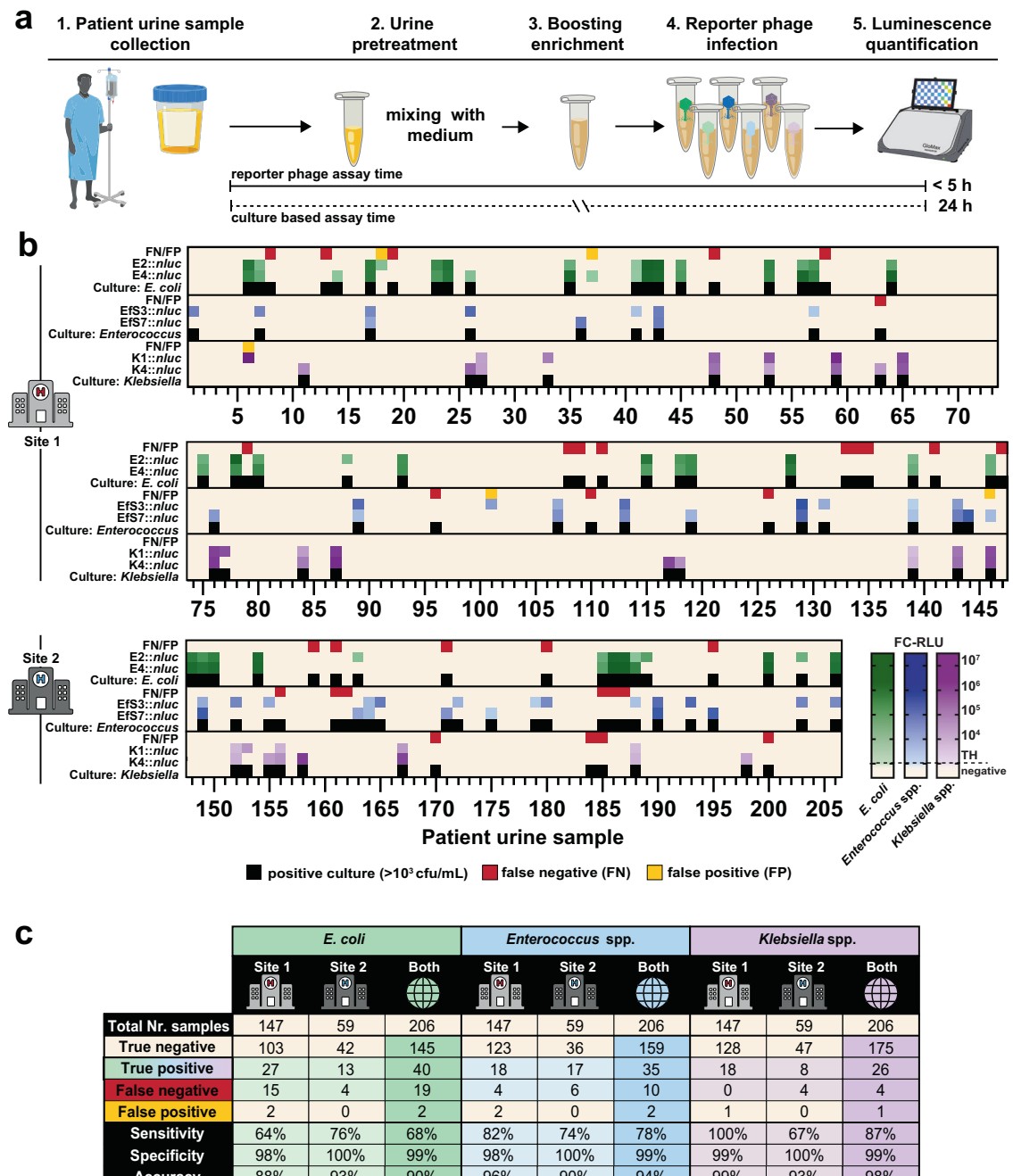

**Fig. 5 | Field evaluation demonstrates rapid, sensitive, and reliable detection of *E. coli*, *Enterococcus* spp. and *Klebsiella* spp. in patient urine.** Workflow of the urine diagnostic assay is shown in **a**. Fresh patient urine was collected and bacterial metabolic activity activated by a short 1 h enrichment in growth media followed by addition of reporter phages. Luminescence was quantified at 3 h (*E. coli*, *Klebsiella* spp.) and 4 h (*Enterococcus* spp.) post infection. **b** Luminescence quantification (RLU) was correlated to bacterial CFU/ml counts from differential/selective plating of 206 patient urine samples on UriSelect and KFS agar plates. In parallel, all results were confirmed by the diagnostic laboratory of the Institute of Medical Microbiology (IMM). Urine samples were derived from two different hospitals (Site 1 and

Site 2). Thresholds (TH) for luminescence were ≥ 100 FC-RLU (*E. coli*, green and *Klebsiella* spp., purple) and 7.5 FC-RLU (*Enterococcus* spp., blue). The test was considered positive if at least one phage per target pathogen had activity >TH. False negatives (FN, red) are samples with no luminescence but bacterial growth (≥ 10³ CFU/ml), false positives (FP, yellow) exhibit luminescence but no bacterial growth. **c** Summary of the test performance showing calculated sensitivity, specificity, accuracy for all target pathogens at both sampling sites. Some elements of the figure were created using biorender.com. Strains from the field evaluation are listed in Table S3. Source data are provided as a Source Data file.

phage-based screening could be implemented as an inclusion criterion for phage therapy trials in the future, depending on the pathogen.

## Discussion

By enabling rapid microbial identification, POC diagnostic platforms (e.g., droplet microfluidics, electrochemical biosensors) have the

potential to assist antimicrobial stewardship and improve patient care[7]. For UTIs, the adaptation of these emerging technologies to direct testing in urine remains challenging, largely owing to their uncontrollable response to complex urine matrices of varying pH and ionic strength[7]. Contrarily, while less complicated technologies (e.g., immunoassays and urine dipsticks) are matrix-insensitive and

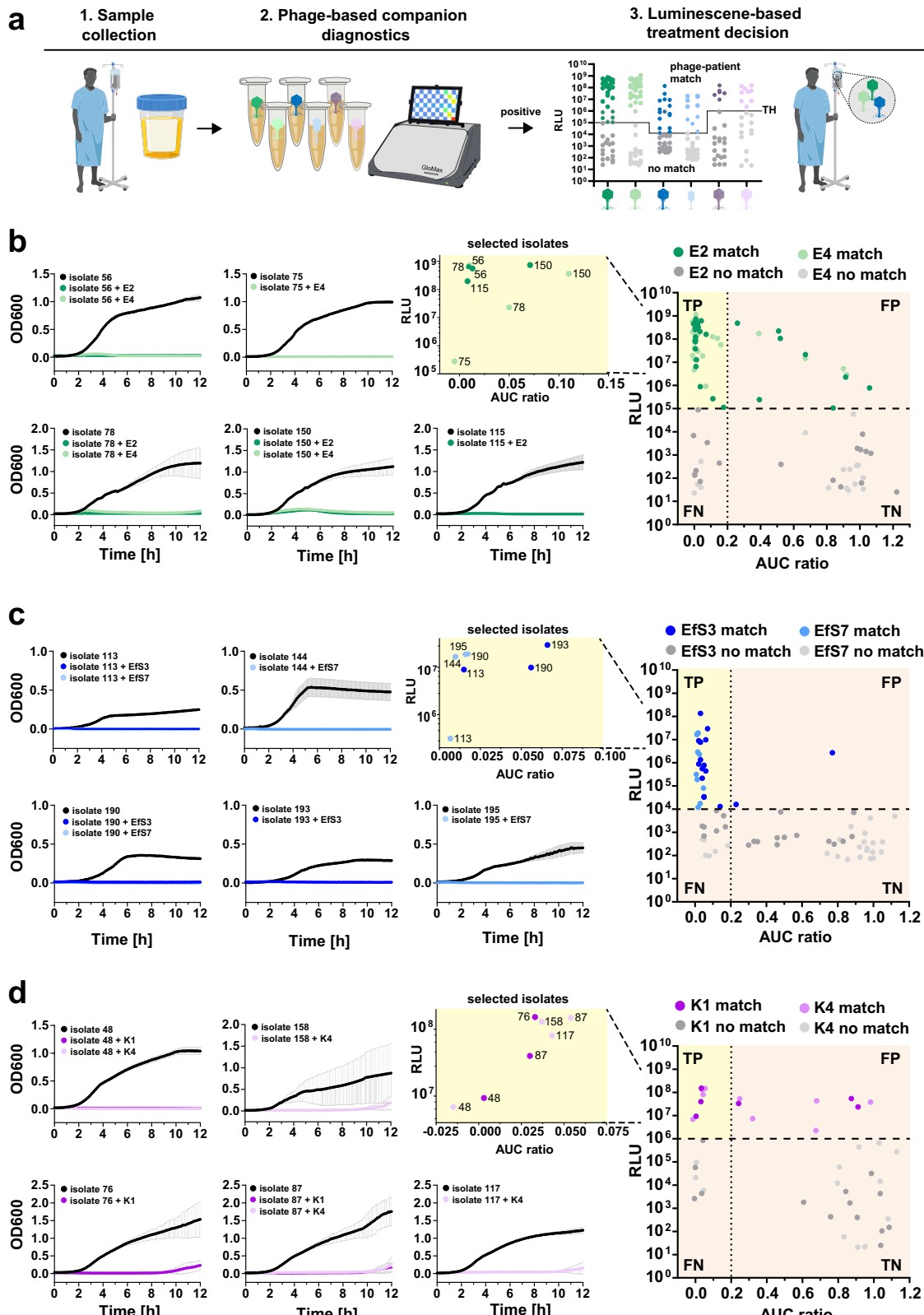

**Fig. 6 | Bioluminescence in patient urine predicts antimicrobial effect of the parental phage scaffold. a** Process of phage-patient matching based on reporter phage-mediated luminescence in urine samples. **b** *E. coli*, **c** *Enterococcus*, and **d** *Klebsiella* strains from the field evaluation were isolated and tested in vitro against each wildtype phage using turbidity reduction assays over 12 h in SHU. Activity was quantified as the ratio of the area under the curve (AUC ratio: AUC$_{HOST+PHAGE}$/AUC$_{HOST}$) where a value of one corresponds to no activity and a value of zero corresponds to maximal activity. For each phage–host pair, the AUC ratio was

plotted against the raw luminescence value from the field evaluation and the thresholds for optimal activity prediction were applied to these dotplots. Optimal RLU and AUC ratio thresholds values were determined as detailed in Fig. S5. For each pathogen, true positives (TP) are highlighted in yellow. Turbidity-reduction curves of selected correctly identified phage-patient matches are displayed on the left to demonstrate host killing. All in vitro data is shown as mean ± SD (*n* = 3). TN true negatives, FP false positives, FN false negatives. Some elements of the figure were created using biorender.com. Source data are provided as a Source Data file.

amenable to POC testing, they often lack adequate accuracy and resolution[7]. In this study, we resolved these two technological dilemmas by developing a reporter phage-based assay. Using *E. coli*, *Klebsiella* spp., and *Enterococcus* spp. as targets, we demonstrated the feasibility of this platform for identifying the predominant uropathogens from unprocessed patient urine at a resolution of ≥10³ CFU/ml within 5 h (Fig. 5). The assay demonstrated high specificity with only few cross-reactivities observed, such as the *Klebsiella* reporter phages detecting *Raoultella*, a close phylogenetic relative of *Klebsiella*[23] and EfS3::*nluc* detecting *Staphylococcus* and *Pediococcus*. However, these false-positives should not impact patient care as *Raoultella* is treated analogous to other Enterobacterales infections (including *Klebsiella*) and staphylococci such as *S. aureus* are rarely found in urine (13/663 urological isolates, Fig. S1). Further efforts are clearly necessary to improve the sensitivity for all three assays (68% for *E. coli*; 78% for *Enterococcus* spp., and 85% for *Klebsiella* spp.), which directly stems from the limited host range of natural phages. This could be addressed by engineering additional reporter phages with complementary host ranges, or by expanding the detection spectrum of the current phages through genetic engineering of phage receptor binding proteins[31–34].

Rapid POC identification and species-level differentiation of bacterial pathogens could greatly assist empiric therapy with antibiotics, and seems particularly useful for treating common infections, such as uncomplicated UTIs. Through early pathogen identification, the treatment regimen can be adjusted ensuring a minimal risk of treatment failure. For example, the increasing prevalence of extended-spectrum beta-lactamase producing Enterobacterales[35] advocates for administering nitrofurantoin, trimethoprim and sulfamethoxazole, or fosfomycin instead of other antibiotics (such as aminopenicillins or cephalosporins) upon POC Enterobacterales identification[36]. These other first-line antibiotics could therefore be reserved for treating other bacterial infections, for example, amoxicillin remains the treatment of choice for *E. faecalis* infections given over 99% of isolates remain susceptible to this antibiotic[37]. At the same time, identification of *Enterococcus* contraindicates the use of cephalosporins due to the intrinsic resistance of enterococci to this class of antibiotics[38]. Moreover, our findings suggest that reporter phages can successfully detect pathogens in whole-blood within less than 6 h, opening promising avenues for septicemia diagnosis. This further supports the adaptability of reporter phage-based diagnostics to diverse and complicated biomatrices.

Despite their transformative potential in bacterial diagnostics, the adoption of reporter phage assays in the clinical realm is still in its infancy. Challenging bottlenecks include the requirements for instrumentation, system automation, workflow integration, and more importantly, establishment of reporter phage libraries covering the clonal diversity of individual pathogens. Our data suggest that while this is achievable for some pathogens (e.g., *E. coli* and *Enterococcus* spp.), it may be more challenging for others such as *Klebsiella* spp., which may in part be due to *Klebsiella* capsule diversity and biofilm formation as inherent barriers to phage infection. Another tangible future for reporter phages is their application as companion diagnostics for phage therapy (Fig. 6). Using bioluminescence as a readout, the susceptibility of a clinical pathogen to therapeutic phage candidates can be quickly screened to determine patient eligibility for phage therapy or to identify suitable phages for a particular patient (personalized therapy). These assays can be performed rapidly (<5 h) in patient matrix without the need to isolate the pathogen before initiating therapy, which would enable the use of phages as "first-line" antimicrobials. In our assay, we demonstrate RLU to be a good predictor of antimicrobial activity for *E. coli* and *Enterococcus* with only few false positive cases (*E. coli*: 12%; *Enterococcus*: 4%). Strains identified as false-positives receive a phage genome but are not killed, likely due to the presence of intracellular resistance[22]. Genetic engineering can be used to target these poor responders through enhancement of phage antimicrobial efficacy, e.g.,

through the delivery of CRISPR-Cas nucleases or other effectors[39]. In this study, we developed technologies to modify our phage candidates and provide the first genome engineering tool for virulent *Enterococcus* phages. Using this pipeline, we have recently created the next generation of phage-based UTI therapeutics that enhance uropathogen killing through phage-mediated delivery of bacteriocins as antimicrobial effectors (heterologous effector phage therapeutics, HEPTs)[40]. Prior to HEPT treatment, E2::*nluc* was used to identify urine samples from responder patients, showcasing how reporter phage diagnostics could be combined with engineered phage therapy in the future to realize the full potential of phage-based precision medicine.

## Methods

### Ethical approval
All patients gave a general written informed consent, in line with the local ethics committee (Kantonale Ethikkommission Zurich, Switzerland), agreeing to further use of health-related personal data and biological material for research purposes. The study was performed in accordance with the World Medical Association Declaration of Helsinki[41] and conforming to the International Conference on Harmonization (ICH) Good Clinical Practice (GCP) Guidelines (E6)[42] and the International Organization for Standardization (ISO, 14,155).

### Bacterial strains and culture conditions
Laboratory strains *E. coli* BL21 (NEB), *K. pneumoniae* KpGe (31) (Dicty Stock Center (32)), and *E. faecalis* JH2-2 (for EfS3) or *E. faecalis* Ef57 (for EfS7) were used as bacteriophage propagation and engineering hosts. *E. coli* strains XL1-Blue MRF′ (Stratagene) or BL21 were used as a cloning host for plasmid construction. Gram-negative bacteria were grown at 37 °C in Luria Bertani (LB) broth. *Enterococcus* spp. and all other Gram-positive isolates except staphylococci were grown at 37 °C in BHI-fc broth or BHI-fc agar plates (37 g/L Brain-Heart-Infusion broth from Biolife Italiana, 4 g/L glycine, 3.2 mM L-cysteine HCl, 50 mM Tris, 5 ng/ml choline chloride) or SHU[16] under microaerophilic conditions. *Staphylococcus* strains were cultured at 37 °C in 0.5 × Brain Heart Infusion (BHI) broth (Biolife Italiana). All clinical strains within the Zurich Uropathogen Collection are compiled in Table S1. Strains used for phage isolation and in vitro functional assessment can be found in Tables S2 and S3. The Zurich Uropathogen Collection is a library of 663 bacterial strains identified and cultured from urine specimens of patients from the Department of Neuro-Urology, Balgrist University Hospital, Zurich, acquired between January and December 2020 and provided after routine testing by the team of the Institute of Medical Microbiology (IMM), University of Zurich. In brief, urine samples were analyzed by selective plating using sheep blood agar (COS, bioMérieux, Marcy l'Etoile, France), sheep blood agar with colistin and nalidixic acid (CNA, bioMérieux) and chromogenic agar (Uriselect 4, Bio-Rad, Marnes-la-Coquette, France). Bacterial growth was determined after 18 h of incubation at 37 °C, 5% $CO_2$. Bacteria were distinguished and identified by MALDI-ToF MS[27]. Data acquisition and analysis were performed by a Maldi Biotyper smart spectrometer (Bruker Daltonics, Bremen, Germany) using the MALDI Biotyper software package (MBT Compass 4.1) using the MBT reference library (version 2021, including 3893 species) and default parameter settings. Antimicrobial susceptibility testing of isolated *E. coli* and *Klebsiella* spp. was performed according to the EUCAST Version 11.0, 2021 (www.eucast.org). All strains are stored at ETH Zurich as glycerol stocks at −80 °C.

### Phage isolation
Bacteriophages were isolated from different wastewater sources in Switzerland using four to five clinical isolates in a sequential host propagation approach[15] (E2: Balgrist Hospital wastewater; E4: EAWAG Dübendorf, Switzerland; EfS3: ARA Wädenswil-Rietliau, Wädenswil, Switzerland; EfS7: CDAM Consorzio Depurazione Acque Mendrisio e Dintorni, Mendrisio, Switzerland). *E. coli* BL21, *K. pneumoniae* KpGe, and

*E. faecalis* JH2-2 were included in the isolations sequence as propagation hosts. Briefly, fresh wastewater from the inlet of wastewater treatment plants were mixed with BHI-fc/LB at a ratio of 1:1 and enriched overnight at 30 °C. Enriched wastewater was cleared by centrifugation (5000 × g) and sterile-filtration (0.22 μm), mixed with a log-phase culture of isolation host A in SHU, pH 6[16], and incubated for 4 h at 30/37 °C. The infection mixture was sterile filtered and subjected to soft-agar overlays using host A. After overnight (ON) incubation at 30 °C, all plaques were collected in 500 μL SM buffer (100 mM NaCl, 8 mM MgSO$_4$ •7 H$_2$O, 50 mM Tris, pH 7.4) and incubated for 2 h at 4 °C. In case of complete lysis, the agar plate was extracted with 5 ml SM buffer. 10–100 μL sterile-filtered supernatants were subsequently mixed with 250 μL SHU and 10 μL log-phase culture of host B and the procedure was repeated. Once a minimum of five hosts had been used for isolation, extracted phages with distinct plaque morphologies were subjected to three rounds of plaque purification on a chosen propagation host P (which was included in the middle of the isolation sequence) and tested for maintained infectivity on all hosts. EfS7 was isolated using five EfS3-resistant hosts, leading to a host-range that is complementary to EfS3. Phage lysates were sterile-filtered (0.22 μm) and stored at 4 °C.

## Phage propagation and purification

**E. coli and Klebsiella spp. phages.** High titer stocks of *E. coli* and *K. pneumoniae* phages were propagated on their respective laboratory hosts in liquid broth. In brief, 100 μL of phages ($10^9$–$10^{10}$ PFU/ml) were added to 500 ml logarithmically growing bacteria (OD$_{600}$ of 0.5) at a multiplicity of infection (MOI) of 0.01–0.1. The infection mixture was incubated at 37 °C with agitation for 5 h or until bacterial clearance. For further purification of all phages, large batch crude lysates were centrifuged at 10,000 × g for 30 min and filter sterilized (0.45 μm) to remove bacterial debris.

**E. faecalis phages.** For *E. faecalis*, the soft-agar overlay methods was used for phage propagation and plaque phenotype determination. BHI-fc soft agar (4% agar, 37 g/L BHI, 4 g/L glycine, 50 mM Tris) supplemented with 3.2 mM L-cysteine HCl was used as top agar layer and BHI-fc agar (1% agar, 37 g/L BHI, 4 g/L glycine, 50 mM Tris, 5 ng/ml choline chloride) served as bottom agar. A total of 200 μl log-phase host bacteria were mixed with 10 μl phage dilution and poured on top of the bottom agar layer. EfS3 and EfS7 were propagated under microaerophilic conditions (Genbox Microaer, biomerieux) at 30 °C. Progeny virions were extracted using 5 ml SM buffer per plate (100 mM NaCl, 8 mM MgSO$_4$, and 50 mM Tris, pH 7.4) and filter-sterilized (0.2 μm) to obtain crude lysates.

**Phage purification.** For further purification, lysates were digested with DNase I (10 μg/ml) and RNase A (1 U/10 ml) at 37 °C for 30 min, phages were concentrated by PEG precipitation (7% PEG 8000 and 1 M NaCl), purified by CsCl isopycnic centrifugation[43], and dialyzed twice against 1000× excess of SM buffer. The purified and concentrated phage stocks were stored at 4 °C.

## Transmission electron microscopy

Phage particles were negatively stained for 20 s with 2% uranyl acetate on carbon-coated copper grids (Quantifoil) and observed at 100 kV on a Hitachi HT 7700 equipped with an AMT XR81B Peltier cooled CCD camera (8 M pixel) at the ScopeM facility, ETH Zurich.

## Phage genome sequencing

CsCl-purified phages were digested with proteinase K (200 μg/ml, 55 °C, 30 min, in SM buffer supplemented with 10 mM EDTA, pH 8.0) and the phage genomic DNA was purified using organic solvents as previously described with two additional purification steps using phenol:chloroform:isoamyl alcohol (25:24:1) followed by DNA precipitation in ethanol[44]. Purified DNA was Illumina sequenced

(2 × 150 bp) by Eurofins Genomics Europe Sequencing GmbH (Constance, Germany). Single contigs were obtained for all six genomes by de novo assembly using the CLC Genomics Workbench version 20 (QIAGEN Bioinformatics) with default settings. A coverage quality analysis performed by aligning genomic reads to individual, assembled contigs revealed no terminal repeat regions in E2, E4, K1, and K4. A 1912 bp terminal redundancy was identified in EfS3 as a region of double coverage. Coding DNA sequence (CDS) identification and annotation was performed using the RAST server[43], with tRNAscan-SE used to confirm tRNA genes[45]. Subsequent manual curation and validation was performed using *E. coli* phage T4 (NC_000866; for E2 and E4) and *Klebsiella* phage JD18 (KT239446; for K1 and K4) as reference genomes.

## Construction of pSelect vectors (CRISPR-Cas9 counterselection)

CRISPR-Cas9 counterselection plasmids (pSelect) were constructed by cloning synthetic spacer sequences into the previously established pCas9 backbone (Addgene no.42876). Under guidance of designated crRNAs, the pCas9 system utilizes a type-IIA CRISPR-Cas9 machinery from *Streptococcus pyogenes* for restriction of targeted genomic sequences[46]. To increase efficiency of phage genome restriction, two spacers targeting two adjacent genes were designed for each phage and introduced into the same plasmid as a repeat-spacer1-repeat-spacer2-repeat (RS$_1$RS$_2$R) unit. The *cas9*, tracrRNA, and crRNA (RSRSR) elements for *E. faecalis* counterselection were amplified from pCas9$_{T4}$ a vector targeting phage T4 that was generated as a control in this study (Table S4) and cloned into the *E. coli-Enterococcus* shuttle vector pLEB579[47]. The resulting vector pSelect$_{T4}$ was used as non-targeting control in *Enterococcus* and is also referred to as scrambled (scr). To construct the counterselection plasmids, the crRNA region of pSelect$_{T4}$ was replaced with a synthetic (GeneArt, ThermoFisher Scientific) RS$_1$RS$_2$R sequence targeting the *cps* regions of EfS3 or EfS7, resulting in the vectors pSelect$_{EfS3}$ and pSelect$_{EfS7}$. Due to the presence of highly conserved genomic sequences across the editing sites of the two phages infecting *E. coli* and *Klebsiella*, commonly applicable pSelect and editing templates were used, i.e., E2 and E4 were engineered with the same set of plasmids, and the same principle applied to K1 and K4. Nucleotide sequences of synthetic DNA strings encoding the CRISPR arrays and primers are summarized in Table S5. All plasmids (Table S4) were constructed using the Gibson isothermal assembly method (NEBuilder HiFi DNA assembly master mix, NEB) and electro-transformed into *E. coli* XL1-Blue MRF′ cells for plasmid production.

## Construction of pEdit vectors (homologous recombination)

Editing plasmids were constructed to mediate sequence-specific integration of the *nluc* gene sequence downstream of the major capsid gene (*cps*) of E2/E4/EfS3/EfS7 or the prohead assembly protein gene (*gp167/gp168*) of K1/K4. Approximately 300–700-bp homology arms flanking the sequence to be inserted were designed to facilitate site-directed recombination. In addition, a single silent nucleotide substitution was introduced on each side of the homology arms within the target PAM sequence (NGG) to enable recombinant phages containing the *nluc* gene to escape CRISPR interference. Homology arms and the synthetic, codon-optimized[48] *nluc* sequence including a strong RBS (5′-GAGGAGGTAAATATAT-3′) were synthesized as gene strings and cloned into editing vectors using pUC19 (pEdit$_E$, pEdit$_K$) or pLEB579 (pEdit$_{EfS3}$ and pEdit$_{EfS7}$) as backbones (Table S4). Sequences of synthetic DNA strings and nucleotide primers are summarized in Table S5.

## Electrotransformation of E. faecalis

*Enterococcus* propagation hosts were grown in osmotically stabilized SGBHI-fc medium with 2% glycine (37 g/L BHI, 20 g/L glycine, 50 mM Tris, 0.5 M Sucrose, 3.3 mM L-cysteine HCl) to log phase growth and

were subsequently washed 2× with 1:10 culture volume Sucrose Glycerol Wash Buffer (SGWB; 0.5 M Sucrose, 10% Glycerol) by centrifugation (5000 × g, 10 min). Electrotransformation occurred at 2.0 kV/cm, 200 Ω, 25 μF. Transformed cells were recovered in SBHI-fc (37 g/L BHI, 0.5 M sucrose, 3.3 mM L-cysteine HCl, 50 mM Tris) for one hour before plasmid selection on BHI-fc agar containing the appropriate antibiotics.

## CRISPR-Cas9-assisted construction of reporter phages

**Enterococcus spp. reporter phages.** For *E. faecalis* phages EfS3 and EfS7, the pEdit vectors were transformed into the propagation hosts of each phage. Subsequently, pEdit-containing strains were infected with phages EfS3, EfS7 using the soft-agar overlay method to obtain semiconfluent lysis and the resulting lysates containing both wildtype and recombinant phage were extracted with 5 ml of SM buffer, sterile-filtered (0.22 μM), and subjected to counterselection on *E. faecalis* strains containing the corresponding pSelect vectors. Multiple plaques were isolated for each phage and correct genotype was assessed by PCR (Primers P19-P22, Table S5) and Sanger sequencing. Individual plaques were extracted and dissolved in 100 μL SM buffer, used as PCR template and directly for luminescence quantification according to the manufacturer's instructions (20 μL phage plus 20 μL substrate, Nano-Glo® Luciferase Assay System, Promega).

**E. coli and Klebsiella spp. reporter phages.** For *E. coli* and *K. pneumoniae*, the CRISPR-Cas9 counterselection plasmid (pSelect) and the corresponding editing plasmid (pEdit) were co-transformed into engineering strains, either *E. coli* BL21 for E2/E4, or *K. pneumoniae* KpGe for K1/K4 by electrotransformation as previously described. Subsequently, the cells were infected with serial dilutions of wildtype phages using the soft agar overlay method to obtain well-isolated lysis plaques. In brief, 10 μL of serially diluted phages was mixed with 200 μL of an overnight bacterial culture in 3 ml of melted LC soft agar (0.4% LB agar supplemented with 10 mM CaCl₂, 2 mM MgSO₄, and 10 g/L glucose) at 47 °C, and subsequently poured onto LB agar plates. After incubation at 37 °C for 12 h, eight candidate plaques were picked for each phage from the infection plates and assayed for correct genotypes using PCR. Of note, due to the low restriction efficiency of selected spacers (average Efficiency of Plating (EOP) > 0.1), all recombinant phages were identified through at least two rounds of plaque screening. To ensure homogeneity of obtained recombinants, insert-positive plaques were further purified through three sequential rounds of plaque purification on their respective spacer-containing hosts (in the absence of pEdit). These successive rounds of CRISPR counterselection enabled elimination of any residual wildtype phages present. Thereafter, the recombinant DNA sequence of purified phages was PCR-amplified and verified by Sanger sequencing (Microsynth AG, Balgach, Switzerland).

## Bioluminescence detection range and specificity assays

Bacterial log-phase cultures were diluted in culture medium/SHU or filtered human urine to an OD₆₀₀ of 0.01 (~10⁷ CFU/ml) and infected with reporter phages at an MOI of 0.1 (*E. coli* and *Klebsiella*) or at an MOI of 2 (*Enterococcus*). Sterile medium spiked with reporter phages alone served as background controls. For E2, E4, K1, and K4, the infection reactions were incubated at 37 °C with shaking (180 rpm) for a total duration of 3 h. For EfS3 and EfS7, the reactions were incubated at 30 °C without shaking for 4 h. For time-course assays, an aliquot of each mixture was taken every 20 min for luminescence measurement in Nunc™ F96 MicroWell™ plates (Thermo Fisher) and using a Glo-Max® navigator luminometer (Promega) with 5 s integration and 2 s delay. Based on manufacturer's instructions, the buffer-reconstituted NLuc substrate (Nano-Glo Luciferase Assay System; Promega) was added to the infection mixtures at 1:1 ratio (40 μL total). Relative light units (RLUs) were divided by the RLU from phage-only controls and

quantified as fold change RLU. All strains used for specificity assays are listed in Table S2.

## Sensitivity and dose-response assays

Stationary-phase bacterial cultures were 10-fold serially diluted with culture medium to a concentration range of 10⁰–10⁷ CFU/ml. 10⁶ PFU/ml (*E. coli* and *Klebsiella*) or 2 × 10⁷ PFU/ml (*Enterococcus*) reporter phages were added to each dilution and incubated at 37 °C with shaking (180 rpm) (*E. coli and Klebsiella*) or at 30 °C without shaking (*Enterococcus*). Luminescence was measured at 3 h (*E. coli* and *Klebsiella*) or 4 h post infection (*Enterococcus*). RLU values were plotted against bacterial log CFUs (determined by plating). The lowest detectable signal (background threshold) was calculated as the average background signal plus three standard deviations (3σ). To determine the bacterial CFU detection limit, a linear regression curve was fitted through the data points above the background threshold, and the intersection between the fitted curve and the background threshold was defined as the detection limit of the assay.

## Dose response assays in whole human blood and urine

O/N cultures of *E. faecalis* were diluted to 10⁸, 10⁶ and 10⁴ CFU/ml in PBS and then spiked separately into whole human blood or filtered human urine to an end concentration of 10⁷, 10⁵, and 10³ CFU/ml. Actual CFU numbers were quantified by plating immediately. After 1 h of boosting enrichment at 37 °C the artificially spiked blood/urine and an unspiked control were mixed with 2 × 10⁷ PFU/ml *Enterococcus* reporter phages EfS3::*nluc*, EfS7::*nluc* in separate tubes. The samples were incubated at 30 °C and luminescence was quantified after 2 h and 4 h post infection (20 μl sample plus 20 μl substrate, Nano-Glo® Luciferase Assay System, Promega). The sample was carefully removed from the upper part of the tube without disturbing the sedimented erythrocytes in the blood samples.

## Phage activity assays

**Spot-on-the-lawn assay.** For *E. coli* and *Klebsiella* phages, 200 μL stationary-phase bacterial culture was mixed with 3 ml of molten LC soft agar and overlaid onto an LB agar plate. Once solidified, serial dilutions (10 μL) of wildtype phages with titer between 10³ and 10⁶ PFU/ml were spotted onto the bacterial lawn, incubated for 18 h at 37 °C, and inspected for plaque formation. For *E. faecalis* phages, 5 ml of a 1:10 diluted log-phase bacterial culture (in BHI-fc) was poured onto a BHI-fc agar plated and incubated for 10 min. Medium was aspirated and the plate dried. Subsequently, phage dilutions were spotted as described above and plates were incubated for 18 h at 30 °C in microaerophilic conditions. Bacterial strains were considered plaque-permissive when plaques formed at any phage dilution. Of note, strains with no plaque formation, but demonstrated growth impairment (i.e., "lysis zones"), were not considered plaque-permissive.

**Liquid infection assay.** Stationary bacterial cultures of *E. coli* and *Klebsiella* were diluted to an OD of 0.1 (10⁷ CFU/ml) in LB medium and infected with phage at an MOI of 0.1 in clear-bottom 96-well plates. In contrast, log-phase *E. faecalis* cultures were diluted in BHI-fc to an OD of 0.1 and infected with 2 × 10⁷ PFU/ml phage. Plates were sealed with a microplate sealing film (Axygen™) and OD₆₀₀ quantified every five minutes at 30 °C for 8 h using a Spectrostar Omega or Spectrostar Nano plate reader (BMG Labtech). Uninfected bacterial cultures and medium were used as growth- and sterility controls, respectively.

**Turbidity reduction assay.** Stationary phase cultures of clinical isolates from the field evaluation (Table S3) grown in SHU were diluted to OD 0.05 in SHU. A total of 100 μl of diluted cultures were mixed with 100 μl parental phage at a therapeutic dose (10⁹ PFU/ml). OD was monitored over 12 h at 30 °C as described. Uninfected bacterial cultures served as growth controls. Curves were background corrected (SHU only) and

the Area Under the Curve was calculated using Graphpad Prism (V. 9.2.0.). From this, the relative reduction in area under the curve (AUC ratio = $AUC_{phage}/AUC_{control}$) was quantified. Correlation of AUC ratios with RLU values of the corresponding reporter phage-clinical isolate combination allowed classification into true positives (TP), true negatives (TN), false positives (FP) and false negatives (FN). Sensitivity (TP/(TP + FN)), Precision (TP/(TP + FP)) and F1-score (2\*TP/(2\*TP + FP + FN)) of the assay were calculated to define the RLU/AUC thresholds that yield the best prediction for antibacterial efficacy.

### Reporter phage-based detection of *E. coli* and *Klebsiella* spp. and *Enterococcus* spp. in patient urine (field evaluation)

From February to June 2021, fresh patient urine was collected at the Balgrist University Hospital, Zurich (147 samples) and the University Hospital Zurich (59 samples) as part of standard practice with routine urine analysis performed by the IMM, with a urine sample procured and stored at 4 °C before testing with the reporter phage assay. Patients included those with a suspected UTI or requiring preoperative urine analysis and featured various forms of bladder emptying (e.g., spontaneous, intermittent catheterization, and indwelling catheter). Fresh urine was mixed with medium (1 ml urine + 4 ml LB for *E. coli* / *Klebsiella* detection; 1 ml urine + 1 ml BHI-fc for *Enterococcus* detection). Preparations were enriched at 37 °C for 1 h with or without (*Enterococcus*) shaking (180 rpm). A total of 500 μL of enriched urine were infected with reporter phages at a final concentration of $10^6$ PFU/ml (E2, E4, K1, and K4 reporter phages) or $10^7$ PFU/ml (EfS3 and EfS7 reporter phages) and incubated as described. Each reporter phage infection was performed in a separate tube. Medium spiked with reporter phages alone served as background controls. Bioluminescence measurements were taken at 3 h (*E. coli*, *Klebsiella*) or 4 h (*Enterococcus*) post infection. FC-RLU values were calculated, urine samples scored, and then compared to the gold standard (GS) results from selective/differential plating.

### Selective/differential plating of patient urine

To benchmark reporter phage results, fresh urine samples were analyzed by plating at ETH Zurich and at the Institute of Medical Microbiology in Zurich (IMM). At ETH, 100 μl fresh urine and several dilutions (1/10; 1/100) were plated on UriSelect4 (Biorad) and KFS (Biolife italiana) agar for uropathogen differentiation and *Enterococcus* selective identification, respectively. Colony formation was evaluated and quantified after 24 h of incubation at 37 °C and 48 h of incubation at 43 °C, respectively. At IMM, urine samples were analysed through selective plating and MALDI-ToF-MS as part of routine urinalysis[27,28,49,50]. ETH plating results with >$10^3$ CFU/ml were considered culture-positive, with KFS overruling UriSelect4. IMM results were considered positive when >$10^4$ CFU/ml were reported.

### Statistics and reproducibility

All information on sample sizes and statistics can be found in the figure legends and the reporting summary. No statistical method was used to predetermine sample size. The experiments were not randomized. The investigators were not blinded to allocation during experiments and outcome assessment. However, during the field evaluation, gold standard results were not available to the investigators during the experiment but became available only after its completion.

### Reporting summary

Further information on research design is available in the Nature Portfolio Reporting Summary linked to this article.

## Data availability

The final annotated genomes are available from the GenBank database: E2 (UTI-E2; [OL870316]), E4 (UTI-E4; [OL870317]), K1 (UTI-K1; [OL870318]), K4 (UTI-K4; [OL870319]), EfS3 (UTI-EfS3; [OL870611]), and EfS7 (UTI-EfS7; [OL870612]). Source data are provided with this paper.

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

## Acknowledgements

We are very thankful to all members of the clinical care teams at the Balgrist University Hospital, Zurich, and the University Hospital Zurich, for urine sample collection during the field evaluation. We are also grateful to all members of the Institute of Medical Microbiology, Zurich for providing clinical isolates that were used for phage isolation, characterization, and the in vitro functional assessment. In addition, we thank the institute of veterinary bacteriology (University of Bern), the national reference center for emerging antibiotic resistance (NARA, University of Fribourg), and the national reference center for enteropathogenic bacteria and *Listeria* (NENT, Zurich) for provision of strains. We thank Prof. Karin Moelling and Dr. Miguelangel Cuenca for scientific advice. The authors acknowledge ScopeM for their support and assistance for transmission electron microscopy of phage particles. We also thank Marco Kipf and Marc Böhler from EAWAG, Dübendorf, Switzerland, Martin Purtschert from AVAU, Aarau, Switzerland, and Daniele Managlia from CDA Mendrisio, Switzerland for providing wastewater used for phage isolation. J.D., L.L., P.P., C.M., J.B., L.H., V.N., J.M., S.Mc, R.Z., T.M.K, M.J.L, S.K., and M.D. were supported by a Sinergia grant (CRSII5_189957) from the Swiss National Science Foundation (SNSF). S.M. and S.K. were supported through an Ambizione grant (PZ00P3_174108) from the SNSF.

## Author contributions

Conceptualization, S.M., J.D., M.J.L., M.D., S.K.,; Methodology, J.D., S.M., S.K., M.D.; Project administration, S.K., M.D.; Supervision, S.K., M.D.; Investigation, J.D., S.M., S.S., S.G., H.K.B., P.P., C.I.M., J.B., L.H., S.Mi., C.G., A.B., S.K., M.D.; Data curation, J.D., S.M., H.K.B., L.L., M.K.H., V.N., S.Mc.,

R.Z., S.K., M.D.; Visualization, S.M., J.D.; Writing—original draft, J.D., S.M., S.K., M.D.; Writing—review and editing, J.D., S.M., H.K.B., L.L., J.K., J.M., S.Mc., T.M.K., M.J.L., S.K., M.D.; Funding acquisition, L.L., T.M.K., M.J.L., S.K., M.D.; Resources, H.K.B., L.L., M.K.H., R.Z., T.M.K., M.J.L., S.K.

## Competing interests

S.K., M.D., and J.B. are employees of Micreos GmbH and M.J.L. is a scientific advisor to Micreos GmbH. The remaining authors declare no competing interests.
