## [Peer Review File · Nature Communications]

REVIEWER COMMENTS

Reviewer #1 (Remarks to the Author):

The authors describe the construction and characterization of reporter phages and their subsequent application for the rapid detection of bacterial pathogens directly from patient samples (urine). In contrast to other reporter phage assays where mostly well described phages are being used, the authors generated reporter phages from novel phages isolated, selected and characterized within this work in order to meet the needs required for broad range pathogen detection (within the target species or genus) and activity in urine. The authors showed that reporter phage assays can be used to detect viable pathogens in patient urine including additional antibiotic susceptibility testing within a couple of hours, which is, to my knowledge, not possible with conventional methods. The manuscript is scientifically sound, very well written and suitable for publication in Nature Communication. I suggest to publish the manuscript after minor revision.

Comments:

97: spp in headline as

145: In which way was the CRISPR-Cas9 system Enterococcus-adapted?

147: I am not sure if “escape mutants” is the right term here, I would rather refer to them as “recombinant nLuc-carrying phages” or so.

149: I would suggest to write “confirm expected reporter gene integration” or similar instead of “functional” as you’ve already proven functionality by luminescence testing.

177: Can the authors explain how the 154 strains were selected? From the 668 strains mentioned in lines 98 – 101, of which 207, 120 and 130 are E.coli, Klebsiella and Enterococcus, respectively (table s1). On which basis was this subset selected? Also, the strain numbers in table s2 do not add up to 154 as I counted 52, 52 and 69 strains of E.coli, Enterococcus and Klebsiella.

194: Why was the antibiotic susceptibility testing only performed on E.coli strains?

196: I guess you meant “patterns” instead of “patters”?

210: “,” instead of “;” after infectivity

Fig. 2:

A bit hard to follow, PAMs are hard to identify on the figure (thin pink stripes) and in the figure legends, the PAM is only mentioned and referred to the figure symbol etc. Might be helpful to explain the procedure a bit more.

How do the authors explain the up to 10 fold difference in luminescence of nluc phage clones?

Have the authors selected the clones with the highest luminescence and sequenced more clones with different luminescence signals? If so, were there any sequence variations detected causing the difference in luminescence output?

Fig. 3:

b: Colors and shapes of the graphs are very hard to differentiate, which is much better in c. The bright dotted LOD lines in d are also hardly visible.

Also, from which type of sample were the authors retrieving the “background signal”? Would be helpful to mention that in the figure legend.

Legend: With SEM you probably mean standard error of the mean? I would suggest to explain that as the other abbreviations.

Figure 4:

a: How can a very strong RLU signal be achieved if the phage does not plaque?

b: Maybe include individual percentage values for the respective phages alongside the combined values.

Figure legend: b: I would suggest to write “individual” instead of “all” reporter phages

Figure legend 5: most of the verbs are in simple present

Figure 6 / S5: I am a bit confused about the RLU values used for comparison with AUC values. In figure legend 6 it is stated that “raw luminescence value from the field evaluation” were used whereas in figure legend 5 “bacterial strains from the field evaluation were isolated and their luminescence response quantified using standardized infection conditions (starting phage titer = 10^7 PFU / ml; starting bacterial concentration = OD0.01). Am I mixing something up here?

General:

I would also include the phage concentration used for all experiments in the figure legends and results as you have done in Fig. Fd-e.

Did the authors determine the capsule types of Klebsiella strains used in the work? As most Klebsiella phages only infect one or a couple of capsule types this might be the reason why 4/12 Klebsiella strains were not detected on site 2. Which capsule types are predominant in Switzerland?

Throughout the work the authors write “Zurich” as well as “Zürich”. I would suggest to use only one spelling.

I suggest to keep the species/genus names for the target pathogens of the reporter phage assay the same throughout the manuscript. For example, sometimes the authors write “Enterococcus spp.” and sometimes “Enterococcus faecalis” which is a bit confusing.

In my opinion the section “Reporter phages as companion diagnostics for phage-patient matching” needs to be explained in more detail. I find the paragraph a bit confusing, especially the section from line 289 to 293. I suggest to explain the following methodology a bit more to make it understandable for a broader audience: “calculations for F1-scores, precision, and sensitivity were used to determine RLU and AUC ratio cut-offs that yield the best overall prediction of antibacterial efficacy”

Reviewer #2 (Remarks to the Author):

Rapid and accurate pathogen identification is important to guarantee a timely and effective therapy and to limit the preventive prescription of broad-range antibiotics. In this study, Meile et al. developed a rapid bacteriophage-based diagnostic assay to detect the most three prevalent pathogens causing urinary tract infections. They isolated and characterized the uropathogen lytic phages and constructed the nanoluciferase reporter phages using CRISPR-Cas9 technology. They then tested the sensitivity, specificity, and accuracy of the reporter phages in hundreds of patient urine samples. Lastly, they show the potential application of reporter phages as companion diagnostics in personalized therapeutic phage applications. Overall, this study is well-designed and the manuscript is well-written, and the methodology is sound with plenty of details. This phage-based detection technology would be useful to support clinical decision-making in the future. My main concern with the paper is the application advantages of reporter phages as companion diagnostics in phage therapy. I have several important areas where I would need clarification (see Major) below to be able to interpret the findings and ultimately the utility of the technology.

Major:

1. The authors did not assess the phage MOI effects on the tests, especially in the field evaluation in patient urine and the companion diagnostic assay. The optimal MOI may further reduce the detection time and improve the resolution or sensitivity.

2. For broad host-range phage isolation in Figure 1a, what's the rationale to use hosts A/B/P/C/D/P selection workflow? Did the authors test more hosts to further broaden the phage host-range since you isolated a bunch of uropathogen?

3. Line 108, "Prior to performing plaque assays, a short enrichment step in synthetic human urine medium (SHU, pH 6) was included for each host to select for phage activity in high urea-containing environments." I noticed that in the field evaluation of reporter phage in patient urine, the authors did not use SHU. Instead, "Prior to phage addition, urine specimens were enriched in growth medium for 1 h to activate the metabolism of stressed bacteria and to reduce possible matrix effects." (Line 242). Why use SHU medium in the phage selection process? The appropriate growth medium is more suitable for phage selection due to the better bacterial activity? Did the author test the effect of urea concentration on phage activity?

4. The authors used a two-step protocol: homologous recombination and CRISPR-Cas9-assisted counter-selection to engineer virulent Enterococcus phages. To be more straightforward and convenient, is it feasible to co-transform pEDIT donor and pSelect into Enterococcus followed by phage infection and recombination? It will significantly improve the recombination efficiency.

5. Line 210, Efs3::nluc non-specifically produced bioluminescence (but no plaques) on 2/12 tested *S. aureus* strains and a single *Pediococcus acidilactici* isolate. The authors claimed that species level specificity is less critical for enterococci because *E. faecium* is very rarely detected in patient urine. However, in the presented data, *E. faecium* is not rarely detected in the patient urine. In figure 4a, at least 52 urological *E. faecium* isolates (similar amount to *E. coli* and *Klebsiella* spp.) were used for testing reporter phage host range and infectivity. Additionally, from figure S1, *E. faecium* consists of ~20% Zurich uropathogen collection.

6. Line 213, "we demonstrated that phage Efs3::nluc is able to infect 12 out of 14 tested *E. faecium* strains, whereas both the plaquing- and detection range of phage Efs7::nluc did not include *E. faecium*".
i) It should be 9 out of 14 based on figure S3a. ii) Why both the plaquing- and detection range of phage Efs7::nluc did not include *E. faecium*? Considering *E. faecium* is the absolute majority of Enterococcus spp. in Zurich uropathogen collection, which genus of Enterococcus can be infected by Efs7::nluc? If this host genus is minor, why chose Efs7::nluc as the detection phage?

7. Line 218, "Infectivity and bioluminescence generation was maintained in 34/36 patient isolate-phage combinations, suggesting urine-dependent matrix effects do not constitute a major hurdle for assay development." No data to support this conclusion in the current manuscript.

8. Line 253, "Interestingly, while all 15 *Klebsiella*-containing samples from Site 1 were

detected, the assay failed to identify 4 out of 12 positive samples from Site 2, possibly due to site-specific difference in circulating clones.” Did the authors validate the sub-species of these failed 4 samples since they simultaneously did conventional culture-based identification combined with mass spectrometry? This information would be useful for the next improvement of the phage host-range.

9. It seems not clear to me the advantages of the application of reporter phage as companion diagnostics for phage therapy. For a clinical pathogen, the isolated potential phages can be quickly and accurately evaluated by the turbidity reduction assay. Why need to construct the corresponding reporter phages which have false positives and negatives? Or use prepared reporter derivatives which may act differently from the therapeutic phages?

10. In the test of reporter phages as companion diagnostics for phage-patient matching, the authors used the defined infection conditions (107 CFU/ml, 109 PFU/ml, SHU). What’s the MOI used here (100?)? And What MOIs were used for the 104/105/106 CFU/ml test?

11. Figure 6 and S5, S5b and 6b-d RLU/AUC ratio figures, why there are some differences between them? What do they tell respectively? There are some false negative dots in K1/K4 RLU/AUC in fig 6d, but not in S5b.

12. The presence of phage defense mechanisms encoded within the host genomes can explain the false positive. What are the possible reasons for the false negative in the test of reporter phages as companion diagnostics?

13. Line 292, “true positives were gated at an AUC ratio <0.2 and at RLU values >105 (*E. coli*), >104 (*E. faecalis*), or >106 (*Klebsiella* spp.)”. How did the authors determine these gate values for true positives?

14. Line 298, “The positive predictive value (i.e., precision) for the companion diagnostic test was calculated at 78 % (*E. coli*), 93 % (*E. faecalis*), and 47 % (*Klebsiella* spp.)”. These percentage numbers are based on which CFU/ml test? How is the sensitivity for the companion diagnostic test? Please clarify.

15. In companion diagnostic assay, the effects of bacterial burden in patient urine on the prediction (precision and sensitivity) should be discussed.

16. As a potential UTI detection technology, it would be useful to discuss the cost of phage-based detection compared with conventional methods. Such as the yield of viable phage

(E2/E4/K1/K4/EfS3/EfS7) from bacterial culture, phage amounts for each sample detection, and luminescence materials/machine, etc.

Minor:

1. Why there are two no-phage controls with different cell-lysis dynamics in figure 2b (middle)?

2. Line 126, “the nluc sequence was integrated alongside a ribosomal binding site (RBS) downstream of a strongly expressed endogenous promoter of phage structural genes.”

Figure 2 legend, “For E. coli and Enterococcus phages, codon optimized nluc (516 bp + RBS) was incorporated immediately downstream of the major capsid gene (cps)”

Which one is correct? downstream of cps promoter or cps gene?

3. Figures 3c and 4d, it would be better to indicate which controls are used in FC-RLU fold change calculation.

4. Line 255, “6 false positives (3 %) were observed across 206 samples and considering all three target pathogens”. Only 5 false positives were labeled in Figure 5b.

5. Figures 6 and S5, for better understanding, i) Show the full names of TP/TN/FP/FN in S5 legend. “true positives (TP), true negatives (TN), false positives (FP) and false negatives (FN)” ; ii) Show TP/TN/FP/FN information in Fig 6 b-d and legend. iii) Show the definition of Sensitivity/ Precision/ F1-score in the S5 legend. “Sensitivity ($TP/(TP+FN)$), Precision ($TP/(TP+FP)$) and F1-score ($2*TP/(2*TP+FP+FN)$) of the assay were calculated to define the RLU/AUC thresholds that yield the best prediction for antibacterial efficacy.” iv) What’s the significance of F1-score? Any reference?

Point-by-point response to reviewers comments

We would like to sincerely thank all reviewers for taking the time to carefully evaluate our study and for their constructive criticism of our work. A detailed point-by-point response can be found below. We feel that these changes have significantly improved the quality and clarity of this manuscript and hope that it is now acceptable for publication.

Comments from Reviewer #1:

The authors describe the construction and characterization of reporter phages and their subsequent application for the rapid detection of bacterial pathogens directly from patient samples (urine). In contrast to other reporter phage assays where mostly well described phages are being used, the authors generated reporter phages from novel phages isolated, selected and characterized within this work in order to meet the needs required for broad range pathogen detection (within the target species or genus) and activity in urine. The authors showed that reporter phage assays can be used to detect viable pathogens in patient urine including additional antibiotic susceptibility testing within a couple of hours, which is, to my knowledge, not possible with conventional methods. The manuscript is scientifically sound, very well written and suitable for publication in Nature Communication. I suggest to publish the manuscript after minor revision.

We would like to thank the reviewer for recognizing the extra effort we invested in identifying suitable phage candidates and the technology development required for modifying non-model phages. Minor comments are addressed below.

Comments:

97: spp in headline as

Thanks for pointing this out. Changed.

145: In which way was the CRISPR-Cas9 system *Enterococcus*-adapted?

We moved the CRISPR-Cas9 cassette into an *E. coli* – *Enterococcus* shuttle vector (pLEB579) and introduced a repeat-spacer cassette with two phage-targeting spacers. We now removed the word “*Enterococcus*-adapted” as we think this is well described in the methods.

147: I am not sure if “escape mutants” is the right term here, I would rather refer to them as “recombinant nLuc-carrying phages” or so.

We changed the statement to “recombinant, CRISPR-resistant phage candidates”. Although the editing penetrance was 100 % in our case, it is always possible that phages become CRISPR-resistant through incorporation of PAM- or protospacer mutations and we want to reflect that possibility in the statement.

149: I would suggest to write “confirm expected reporter gene integration” or similar instead of “functional” as you’ve already proven functionality by luminescence testing.

Good suggestion. We incorporated this change.

177: Can the authors explain how the 154 strains were selected? From the 668 strains mentioned in lines 98 – 101, of which 207, 120 and 130 are *E.coli*, *Klebsiella* and *Enterococcus*, respectively (table

s1). On which basis was this subset selected? Also, the strain numbers in table s2 do not add up to 154 as I counted 52, 52 and 69 strains of E.coli, Enterococcus and Klebsiella.

These strains were randomly selected and we have integrated this change in the revised manuscript: “a panel of 154 randomly selected urological isolates (52 E. coli, 50 Klebsiella spp., and 52 E. faecalis).” Regarding the strains in the supplemental table: Table S2 does not only contain strains used for host-range / bioluminescence assays, but also strains used for specificity testing, cloning, and phage propagation. To clarify, we now added the use of these strains in Table S2.

194: Why was the antibiotic susceptibility testing only performed on E.coli strains?

According to CLSI guidelines, the diagnostic lab in Zurich (IMM) does not routinely test sensitivity to a broad range of antibiotics when standard-of-care antibiotics are effective. Therefore, we did not have this data for most *Klebsiella* and *Enterococcus* strains.

196: I guess you meant “patterns” instead of “patters”?

Thank you for spotting this. Changed!

210: “,” instead of “;” after infectivity

Thank you. Changed.

Fig. 2:

A bit hard to follow, PAMs are hard to identify on the figure (thin pink stripes) and in the figure legends, the PAM is only mentioned and referred to the figure symbol etc. Might be helpful to explain the procedure a bit more.

We agree that this was a bit confusing. We have implemented changes in the figure to increase clarity. In addition, we now provide a schematic overview of the engineering process in the supplemental material (new: **Figure S2a**, see below):

Fig S2a: Schematic overview of the engineering workflow.

How do the authors explain the up to 10 fold difference in luminescence of nluc phage clones?

Good spot. The bioluminescence shown in this panel was measured from a single picked phage plaque. Cytoplasmic content (including NLuc protein) in the plaque was extracted with 100 μ L SM-buffer and subsequently mixed with substrate. There is typically a high variability due to the picking process. Therefore, the observed differences in luminescence are due to the amount of material picked, not the genetic background.

To clarify we added: “and RLU determination from individual plaques, respectively.” To the legend of figure 2.

Have the authors selected the clones with the highest luminescence and sequenced more clones with different luminescence signals? If so, were there any sequence variations detected causing the difference in luminescence output.

See explanation above. The selected clones were sequence verified and contained no mutations. The reason we chose to show data for 10 clones was to demonstrate the efficiency of the engineering pipeline (100 %).

Fig. 3:

b: Colors and shapes of the graphs are very hard to differentiate, which is much better in c. The bright dotted LOD lines in d are also hardly visible.

We have increased the thickness of the LOD and background luminescence lines. However, to make graphs look like panel (c), we would have to artificially remove datapoints, which we prefer not to do.

Also, from which type of sample were the authors retrieving the “background signal”? Would be helpful to mention that in the figure legend.

Thank you for pointing this out, the background was determined from a sample lacking bacteria, i.e., phage in medium. We have changed the legend to:

“background luminescence (phage in medium, indicated as horizontal dotted lines).”

Legend: With SEM you probably mean standard error of the mean? I would suggest to explain that as the other abbreviations.

Correct, thank you for spotting this, we now introduce this abbreviation in the legend of Fig. 3.

Figure 4:

a: How can a very strong RLU signal be achieved if the phage does not plaque?

This is an interesting question. It will happen whenever a phage can bind to the host cell and inject its genomic DNA but fails to complete a full infection cycle. This is typically due to bacterial intracellular defence systems, such as CRISPR-Cas, RM-systems, or abortive infection. Phage defence is a “hot” research topic and many systems have recently been described (see: review).

We believe that this is addressed sufficiently in lines 179-182 and lines 187-190:

179-182:

“While bioluminescence demonstrates the ability of a phage to inject genetic material into the bacterial cytoplasm and express luciferase, the criterion for plaquing is more stringent, as the phage must complete a full infection cycle and lyse the bacterial cell”

187-190:

“All phages presented significantly broader bioluminescence detection ranges compared to their plaquing ability (Fig. 4b), which could be due to bacterial intracellular defence systems that prevent the completion of a full infection cycle in some strains.”

b: Maybe include individual percentage values for the respective phages alongside the combined values.

Good suggestion, we have included the plaquing and bioluminescence information for each individual phage in Figure 4b.

Bacterial target	E. coli (n=52)			E. faecalis (n=52)			Klebsiella spp. (n=50)		
	E2::nluc	E4::nluc	combined	EfS3::nluc	EfS7::nluc	combined	K1::nluc	K4::nluc	combined
Phage									
Plaquing host range	71 %	58 %	71 %	52 %	33 %	77 %	48 %	54 %	60 %
Detection range	81 %	81 %	83 %	73 %	48 %	96 %	84 %	80 %	90 %

Figure 4b: Summary of individual and combined host- and detection ranges. Compared to the plaquing host range, the detection range was larger for each reporter phage and phage combination.

Figure legend: b: I would suggest to write “individual” instead of “all” reporter phages

Thank you for this comment. We agree that the whole sentence was a bit confusing. We changed it to:

“(b) Summary of individual and combined host ranges. Each reporter phage was able to transduce bioluminescence into strains that were not permissive to plaque formation, effectively leading to an increased detection range compared to the plaquing host range.”

Figure legend 5: most of the verbs are in simple present

Thank you for spotting this. We have changed the tense in the revised manuscript.

Figure 6 / S5: I am a bit confused about the RLU values used for comparison with AUC values. In figure legend 6 it is stated that “raw luminescence value from the field evaluation” were used whereas in figure legend 5 “bacterial strains from the field evaluation were isolated and their luminescence response quantified using standardized infection conditions (starting phage titer = 10⁷ PFU / ml; starting bacterial concentration = OD0.01). Am I mixing something up here?

Thank you for flagging this. You are not mixing anything up, we agree that the text in this section could be improved to better explain the rationale. In Fig. S5 the luminescence was quantified under standardized conditions (10⁷ CFU/ml, 10⁹ PFU/ml, in synthetic human urine) and compared to the AUC ratio. This allowed use to determine the cut-offs that yield the best activity prediction (Fig. S5).

In a real clinical setting, however, one would work with a ill-defined medium (the patient’s urine) and an unknown cell number. To test whether the companion diagnostic would produce a useful activity

prediction, we applied the gating parameters to the raw luminescence values measured during the field evaluation, and this is what's shown in Fig. 6.

We have added the following sentences (new line 296-300):

“The gating parameters (Fig. S5) were determined in a defined growth medium (SHU) with a defined cell concentration (10^7 CFU/ml), which does not reflect a clinical scenario where the patient's urine composition and bacterial cell numbers are unknown. To test how well treatment success can be predicted based only on the raw luminescence values obtained from patient urine, we applied these gating criteria to the data from the field evaluation (workflow: Fig. 6a).”

General:

I would also include the phage concentration used for all experiments in the figure legends and results as you have done in Fig. Fd-e.

We agree that providing this information is helpful in the figure legend. Unfortunately, the conditions differ between Gram-negative and Gram-positive host organisms, which would make the figure legend hard to read. We would therefore prefer keeping the information in the methods sections.

Did the authors determine the capsule types of *Klebsiella* strains used in the work? As most *Klebsiella* phages only infect one or a couple of capsule types this might be the reason why 4/12 *Klebsiella* strains were not detected on site 2. Which capsule types are predominant in Switzerland?

We agree that serotypes and capsule types are important markers for epidemiological studies. Capsule-specific *Klebsiella* phages typically have a very narrow host-range due to the use of enzymatic tail spikes as their primary RBPs. However, we specifically selected phages that use tail fibers with broad binding ranges (K1, K4) which are highly unlikely to rely solely on capsular polysaccharides for host recognition. Furthermore, both phages K1 and K4 belong to the *Jiaodavirus* genus that are related to phage T4 at the family level. Such phages strictly employ tail fibers as their RBPs that also typically recognize different outer-membrane proteins and/or lipopolysaccharide core structures as receptors. We have also begun to characterize the RBPs of all phages used in this work for publication in the future. No tailspike or capsule depolymerase-like genes could be identified within the genomes of K1 or K4 with both phages identified as carrying similar sets of fiber genes as well-described for phage T4 genome, i.e., a short tail fiber gene and a cassette of genes that assemble to form the long tail fiber (K1 LTF: gp242 (proximal subunit) to gp246 (tail adhesin); K4 LTF: gp239 (proximal subunit) to gp243 (tail fiber assembly protein)). Importantly, the distal receptor-binding subunits of the K1 and K4 LTFs are completely different. While K1-gp246 forms the terminal adhesin that would self-assemble onto the fiber tip of gp245 (akin to PMID30244968) and confer receptor binding, the product of gp243 from phage K4 instead functions as a tail fiber assembly (Tfa) protein to help with the folding and maturation of a homotrimeric assembly of gp242 as the terminal fiber that mediates host recognition. K4-gp243 is also structurally distinct from K1-gp246 as it presents high structural similarity to other Tfa proteins such as that of phage Mu (PMID: 31209305).

Figure. AlphaFold prediction of the K1 adhesin (gp246) which forms a unique adhesin structure (*unpublished work*) with an N-terminal (blue) attachment domain that closely resembles that of other fiber adhesins such as that of *Salmonella* phage S16 (PMID: 30244968).

For the reasons mentioned above, we did not determine the capsule type via whole genome sequencing. However, during the revision, we used a recently optimized MALDI-TOF-MS method (PMID: 34517886) to determine the subspecies of the *Klebsiella* strains isolated during the field evaluation to test whether false-negatives may be associated with a particular *Klebsiella* species/subspecies. The summary can be found below.

Table: Subspecies identification of *Klebsiella* isolates from the field evaluation. N.i. = non identified, TP= true positive, FN = false negative, BAL = site 1, USZ = site 2.

Klebsiella species-complex designation	Subspecies designation	Field evaluation result	Patient Nr. field evaluation	Strain designation
Klebsiella aerogenes	n.i	TP	158	USZ Ka11
Klebsiella aerogenes	n.i	FN	200	USZ Ka53
Klebsiella oxytoca	K. oxytoca	TP	27	BAL Ko1
Klebsiella oxytoca	K. oxytoca	TP	48	BAL Ko2
Klebsiella oxytoca	K. michiganensis or K. pasteurii or K. grimontii	TP	65	BAL Ko3
Klebsiella oxytoca	K. michiganensis or K. pasteurii or K. grimontii	TP	139	BAL Ko4
Klebsiella oxytoca	K. pasteurii	TP	156	USZ Ko09
Klebsiella oxytoca	K. oxytoca	TP	198	USZ Ko51
Klebsiella pneumoniae	K. pneumoniae sensu stricto	TP	11	BAL Kp1
Klebsiella pneumoniae	K. pneumoniae	TP	26	BAL Kp2
Klebsiella pneumoniae	K. pneumoniae	TP	76	BAL Kp5
Klebsiella pneumoniae	K. pneumoniae	TP	77	BAL Kp6
Klebsiella pneumoniae	K. variicola subsp. variicola	TP	84	BAL Kp7
Klebsiella pneumoniae	K. quasipneumoniae subsp. similipneumoniae	TP	87	BAL Kp8
Klebsiella pneumoniae	K. pneumoniae sensu stricto	TP	117	BAL Kp9
Klebsiella pneumoniae	K. pneumoniae sensu stricto	TP	188	BAL Kp10
Klebsiella pneumoniae	K. variicola subsp. variicola	TP	143	BAL Kp11
Klebsiella pneumoniae	K. variicola subsp. variicola	TP	146	BAL Kp12
Klebsiella pneumoniae	K. pneumoniae sensu stricto	TP	155	USZ Kp08
Klebsiella pneumoniae	K. pneumoniae sensu stricto	FN	184	USZ Kp37
Klebsiella pneumoniae	K. pneumoniae sensu stricto	FN	185	USZ Kp38
Klebsiella pneumoniae	K. pneumoniae sensu stricto	TP	188	USZ Kp41

In sum, the false-negative identifications from site 2 were *K. aerogenes* (200) and *K. pneumoniae sensu stricto* (patient 184 and 185). We could not grow the fourth false-negative strain (patient 170, *K. pneumoniae* species complex) and thus cannot provide subspecies identification.

Because both *K. aerogenes* and *K. pneumoniae* were identified as true positives from site 1, we conclude that subspecies-specificity does not explain the higher false-negative rate at site 2. No changes were made to the text as we think this is beyond the scope of the current manuscript.

Throughout the work the authors write “Zurich” as well as “Zürich”. I would suggest to use only one spelling.

We have replaced “Zürich” with “Zurich” throughout the manuscript.

I suggest to keep the species/genus names for the target pathogens of the reporter phage assay the same throughout the manuscript. For example, sometimes the authors write “Enterococcus spp.” and sometimes “Enterococcus faecalis” which is a bit confusing.

Thank you for pointing this out. This stems from the fact that Efs3::*nluc* is able to detect *E. faecium* as well as *E. faecalis*. However, *E. faecium* is very rare in urine (1/130 *Enterococcus* isolates in the Zurich Uropathogen Collection) and 1/44 isolated *Enterococcus* spp in our field evaluation)

We have changed to *Enterococcus* spp., wherever appropriate.

In my opinion the section “Reporter phages as companion diagnostics for phage-patient matching” needs to be explained in more detail. I find the paragraph a bit confusing, especially the section from line 289 to 293. I suggest to explain the following methodology a bit more to make it understandable for a broader audience: “calculations for F1-scores, precision, and sensitivity were used to determine RLU and AUC ratio cut-offs that yield the best overall prediction of antibacterial efficacy”

Thank you for this suggestion. We have made several changes to this section to increase clarity. With respect to the F1 score, we provide a high-level description of its use in statistics.

Line 289-293:

“...and calculations for precision, sensitivity and F1-score were used to determine RLU and AUC ratio cut-offs that yield the best overall prediction of antibacterial efficacy for individual phage-host pairs (Fig. S5c). The F1-score is the harmonic mean of precision and sensitivity and can be used to optimize trade-offs between these two metrics. Based...”

Comments from Reviewer #2:

Rapid and accurate pathogen identification is important to guarantee a timely and effective therapy and to limit the preventive prescription of broad-range antibiotics. In this study, Meile et al. developed a rapid bacteriophage-based diagnostic assay to detect the most three prevalent pathogens causing urinary tract infections. They isolated and characterized the uropathogen lytic phages and constructed the nanoluciferase reporter phages using CRISPR-Cas9 technology. They then tested the sensitivity, specificity, and accuracy of the reporter phages in hundreds of patient urine samples. Lastly, they show the potential application of reporter phages as companion diagnostics in personalized therapeutic phage applications. Overall, this study is well-designed and the manuscript is well-written, and the methodology is sound with plenty of details. This phage-based detection technology would be useful to support clinical decision-making in the future. My main concern with the paper is the application advantages of reporter phages as companion diagnostics in phage therapy. I have several important areas where I would need clarification (see Major) below to be able to interpret the findings and ultimately the utility of the technology.

We would like to thank the reviewer for the positive assessment of our work.

Major:

1. The authors did not assess the phage MOI effects on the tests, especially in the field evaluation in patient urine and the companion diagnostic assay. The optimal MOI may further reduce the detection time and improve the resolution or sensitivity.

Thank you for pointing this out. Indeed, the MOI has a strong impact on the dynamics of infection and consequently the detection times. We have previously optimized phage concentrations for other reporter phages (see PMID32245761 and PMID32858938) and observed that higher MOI generally reduces detection time (as can be expected). However, that maximal concentration of reporter phage that can be used is typically limited by carry-over of residual NLuc protein in the phage preparation. This results in background luminescence that can limit the sensitivity of the assay. Very low MOIs on

the other hand can lead to significantly longer detection times (see study by Braun et al.; PMID: 36719711).

Therefore, we always use the highest concentration of reporter phage (minimum of 1×10^6 PFU/ml, depending on the phage) that can be used without producing too much background luminescence. In light of the experience from previous literature, we believe this approach is justified.

2. For broad host-range phage isolation in Figure 1a, what's the rationale to use hosts A/B/P/C/D/P selection workflow? Did the authors test more hosts to further broaden the phage host-range since you isolated a bunch of uropathogen?

The main reason to use such a sequential multi-host isolation method is to systematically exclude phages with very narrow host range. Upon completion of the isolation "pathway", many individual plaques were clonally isolated and tested against a broad range of clinical isolates to prioritize the best candidates. A comprehensive host-range analysis for the selected phages is shown in Figure 4. For the three pathogens, we obtained a combined detection range covering 83 – 96 % of clinical isolates, which corroborates this approach.

3. Line 108, "Prior to performing plaque assays, a short enrichment step in synthetic human urine medium (SHU, pH 6) was included for each host to select for phage activity in high urea-containing environments." I noticed that in the field evaluation of reporter phage in patient urine, the authors did not use SHU. Instead, "Prior to phage addition, urine specimens were enriched in growth medium for 1 h to activate the metabolism of stressed bacteria and to reduce possible matrix effects." (Line 242). Why use SHU medium in the phage selection process? The appropriate growth medium is more suitable for phage selection due to the better bacterial activity? Did the author test the effect of urea concentration on phage activity?

The phages were isolated in SHU to make sure that they remain active in this environment. This is important (I) because patient urine still comprises a significant fraction of the environment in the field evaluation assay and (II) because these phages were isolated as diagnostic AND therapeutic candidate (i.e., they should be active in patient urine). Their therapeutic potential is further explored in a recent preprint (link).

We did not test the effect of changing urea concentrations in this manuscript, but we know that the phages are active in high-urea environments. For example, all the killing data used in Fig. S5 and Fig. 6 were obtained in SHU (250 mM urea).

4. The authors used a two-step protocol: homologous recombination and CRISPR-Cas9-assisted counter-selection to engineer virulent Enterococcus phages. To be more straightforward and convenient, is it feasible to co-transform pEDIT donor and pSelect into Enterococcus followed by phage infection and recombination? It will significantly improve the recombination efficiency.

Yes, it is possible to do recombination and counter-selection in a one-step reaction, although it only saves one day of work. We used such an approach to modify the *E. coli* and *Klebsiella* phages (see methods section) and applied it previously for the modification of *Listeria* phages (PMID: 30053228). For *Enterococcus*, we were limited by the availability of plasmids and therefore opted for a two-step strategy. Our editing efficiency was 100 %, so the editing efficiency cannot be further improved.

5. Line 210, Efs3::nluc non-specifically produced bioluminescence (but no plaques) on 2/12 tested *S. aureus* strains and a single *Pediococcus acidilactici* isolate. The authors claimed that species level specificity is less critical for enterococci because *E. faecium* is very rarely detected in patient urine.

However, in the presented data, *E. faecium* is not rarely detected in the patient urine. In figure 4a, at least 52 urological *E. faecium* isolates (similar amount to *E. coli* and *Klebsiella* spp.) were used for testing reporter phage host range and infectivity. Additionally, from figure S1, *E. faecium* consists of ~20% Zurich uropathogen collection.

We believe that the reviewer is confusing *E. faecium* with *E. faecalis*. As shown in Figure S1, the vast majority of urological *Enterococcus* isolates are *E. faecalis* (125/130), whereas we only identified a single *E. faecium* strain within the whole Zurich Uropathogen Collection.

6. Line 213, “we demonstrated that phage EfS3::nluc is able to infect 12 out of 14 tested *E. faecium* strains, whereas both the plaquing- and detection range of phage EfS7::nluc did not include *E. faecium*”. i) It should be 9 out of 14 based on figure S3a.

Thanks for pointing out this mistake. We have changed the text.

ii) Why both the plaquing- and detection range of phage EfS7::nluc did not include *E. faecium*? Considering *E. faecium* is the absolute majority of *Enterococcus* spp. in Zurich uropathogen collection, which genus of *Enterococcus* can be infected by EfS7::nluc? If this host genus is minor, why chose EfS7::nluc as the detection phage?

See response to comment 5. The vast majority of *Enterococcus* UTIs are *E. faecalis*, not *E. faecium* and EfS7::nluc is specific for *E. faecalis*.

7. Line 218, “Infectivity and bioluminescence generation was maintained in 34/36 patient isolate-phage combinations, suggesting urine-dependent matrix effects do not constitute a major hurdle for assay development.” No data to support this conclusion in the current manuscript.

Thank you for pointing this out. We have moved the figure reference to point the readers towards the data. We believe that there is sufficient data to support this conclusion. We have tested 36 phage/bacteria combinations in pooled human urine. If there were major effects curbing infectivity in urine, this would have become evident.

8. Line 253, “Interestingly, while all 15 *Klebsiella*-containing samples from Site 1 were detected, the assay failed to identify 4 out of 12 positive samples from Site 2, possibly due to site-specific difference in circulating clones.” Did the authors validate the sub-species of these failed 4 samples since they simultaneously did conventional culture-based identification combined with mass spectrometry? This information would be useful for the next improvement of the phage host-range.

We would like to thank the reviewer for this suggestion. During the revision, we used a recently optimized MALDI-TOF-MS method to determine the subspecies of the *Klebsiella* strains isolated during the field evaluation to test whether false-negatives may be associated with a particular *Klebsiella* species/subspecies. The summary can be found below.

Table: Subspecies identification of *Klebsiella* isolates from the field evaluation. N.i. = non identified, TP= true positive, FN = false negative, BAL = site 1, USZ = site 2.

Klebsiella species-complex designation	Subspecies designation	Field evaluation result	Patient Nr. field evaluation	Strain designation
Klebsiella aerogenes	n.i	TP	158	USZ Ka11
Klebsiella aerogenes	n.i	FN	200	USZ Ka53
Klebsiella oxytoca	K. oxytoca	TP	27	BAL Ko1
Klebsiella oxytoca	K. oxytoca	TP	48	BAL Ko2
Klebsiella oxytoca	K. michiganensis or K. pasteurii or K. grimontii	TP	65	BAL Ko3
Klebsiella oxytoca	K. michiganensis or K. pasteurii or K. grimontii	TP	139	BAL Ko4
Klebsiella oxytoca	K. pasteurii	TP	156	USZ Ko09
Klebsiella oxytoca	K. oxytoca	TP	198	USZ Ko51
Klebsiella pneumoniae	K. pneumoniae sensu stricto	TP	11	BAL Kp1
Klebsiella pneumoniae	K. pneumoniae	TP	26	BAL Kp2
Klebsiella pneumoniae	K. pneumoniae	TP	76	BAL Kp5
Klebsiella pneumoniae	K. pneumoniae	TP	77	BAL Kp6
Klebsiella pneumoniae	K. variicola subsp. variicola	TP	84	BAL Kp7
Klebsiella pneumoniae	K. quasipneumoniae subsp. similipneumoniae	TP	87	BAL Kp8
Klebsiella pneumoniae	K. pneumoniae sensu stricto	TP	117	BAL Kp9
Klebsiella pneumoniae	K. pneumoniae sensu stricto	TP	188	BAL Kp10
Klebsiella pneumoniae	K. variicola subsp. variicola	TP	143	BAL Kp11
Klebsiella pneumoniae	K. variicola subsp. variicola	TP	146	BAL Kp12
Klebsiella pneumoniae	K. pneumoniae sensu stricto	TP	155	USZ Kp08
Klebsiella pneumoniae	K. pneumoniae sensu stricto	FN	184	USZ Kp37
Klebsiella pneumoniae	K. pneumoniae sensu stricto	FN	185	USZ Kp38
Klebsiella pneumoniae	K. pneumoniae sensu stricto	TP	188	USZ Kp41

In sum, the false-negative identifications from site 2 were *K. aerogenes* (200) and *K. pneumoniae sensu stricto* (patient 184 and 185). We could not grow the fourth false-negative strain (patient 170, *K. pneumoniae* species complex) and thus cannot provide subspecies identification.

Because both *K. aerogenes* and *K. pneumoniae* were identified as true positives from site 1, we conclude that subspecies-specificity does not explain the higher false-negative rate at site 2. No changes were made to the text as we think this is beyond the scope of the current manuscript.

9. It seems not clear to me the advantages of the application of reporter phage as companion diagnostics for phage therapy. For a clinical pathogen, the isolated potential phages can be quickly and accurately evaluated by the turbidity reduction assay. Why need to construct the corresponding reporter phages which have false positives and negatives? Or use prepared reporter derivatives which may act differently from the therapeutic phages?

We agree that within the current framework of phage therapy (which is mostly compassionate use), a reporter phage-based companion diagnostic will likely not be implemented yet. However, there are several advantages that could drive its implementation in the future:

1. Reporter phage assays can be performed directly in patient urine (or other patient matrix, e.g blood), which is not possible for turbidity reduction assays. Because this process is fast, it enables phage selection on the same day.
2. No isolation of the pathogen is required prior to starting therapy.
3. The assay could be performed in high-throughput, for example using a more extensive library of potential therapeutic phages (future scenario).
4. Bioluminescence is a marker for successful genome delivery, which could be used for the selection of engineered phages that deliver therapeutic payload proteins (some of the phages in this study have been further developed as engineered therapeutic candidates, see BioRxiv manuscript).

We show that our phages do not act differently than their wildtype counterparts (see Fig. 3a-b).

In the revised manuscript, we have expanded the discussion to include some of these considerations:

Line 361-367:

“Another tangible future for reporter phages is their application as companion diagnostics for phage therapy (Fig. 6). Using bioluminescence as a readout, the susceptibility of a clinical pathogen to therapeutic phage candidates can be quickly screened to determine patient eligibility for phage therapy or to identify suitable phages for a particular patient (personalized therapy). These assays can be performed rapidly (< 5h) in patient matrix without the need to isolate the pathogen before initiating therapy, which would enable the use of phages as “first-line” antimicrobials.”

10. In the test of reporter phages as companion diagnostics for phage-patient matching, the authors used the defined infection conditions (107 CFU/ml, 109 PFU/ml, SHU). What’s the MOI used here (100)?

We always infect with 10^9 PFU/ml. This corresponds to an MOI of 100 under defined conditions. In patient urine, the MOI may vary depending on the bacterial load.

And What MOIs were used for the 104/105/106 CFU/ml test?

We believe the reviewer is referring to the RLU cut-offs that were tested in Fig. S5. These values were used to establish gating parameter for the companion diagnostic. To avoid confusion for the reader we changed RLU to bioluminescence in the following sentence:

“Based on these parameter optimizations, true positives were gated at an AUC ratio <0.2 and at bioluminescence values $>10^5$ RLU (E. coli), $>10^4$ RLU (E. faecalis), or $>10^6$ RLU (Klebsiella spp.)”

11. Figure 6 and S5, S5b and 6b-d RLU/AUC ratio figures, why there are some differences between them? What do they tell respectively? There are some false negative dots in K1/K4 RLU/AUC in fig 6d, but not in S5b.

Both figures show a correlation between the amount of luminescence (RLU) and the killing (AUC ratio) mediated by our phages. In Figure S5 the infection and bioluminescence assays were performed under standardized conditions in SHU to determine the cut-off values for gating into TP/FP/TN and FN. Because the number of bacteria (and thus the bioluminescence) in the actual patient urine can vary greatly, we have tested our gating parameters on the bioluminescence values obtained from the field evaluation. This is what is shown in Figure 6. The additional FN samples observed there are likely due to low bacterial numbers.

We have changed this section to increase clarity.

12. The presence of phage defense mechanisms encoded within the host genomes can explain the false positive. What are the possible reasons for the false negative in the test of reporter phages as companion diagnostics?

False negatives: Can be due to low cell numbers in patient urine. We used rather stringent gating criteria to avoid false-positives at the cost of accepting a few false-negatives. This trade off is optimized through F1-score analysis, which is the harmonic mean of positive predictive value and sensitivity (we have added a sentence to clarify this in the manuscript).

Line 289-293:

“...and calculations for precision, sensitivity and F1-score were used to determine RLU and AUC ratio cut-offs that yield the best overall prediction of antibacterial efficacy for individual phage-host pairs (Fig. S5c). The F1-score is the harmonic mean of precision and sensitivity and can be used to optimize trade-offs between these two metrics. Based...”

13. Line 292, “true positives were gated at an AUC ratio <0.2 and at RLU values >105 (*E. coli*), >104 (*E. faecalis*), or >106 (*Klebsiella* spp.)”. How did the authors determine these gate values for true positives?

See response to comment 11. Gating parameters were determined as shown in Fig. S5.

14. Line 298, “The positive predictive value (i.e., precision) for the companion diagnostic test was calculated at 78 % (*E. coli*), 93 % (*E. faecalis*), and 47 % (*Klebsiella* spp.)”. These percentage numbers are based on which CFU/ml test? How is the sensitivity for the companion diagnostic test? Please clarify.

Values were deduced from analysis of the actual patient samples from the field evaluation, which present varying CFU/mL concentrations in their urine.

We changed text to (line 303-305):

“The positive predictive value (i.e., precision) for the companion diagnostic test in patient urine was calculated at 78 % (*E. coli*), 93 % (*Enterococcus* spp., *faecalis*), and 47 % (*Klebsiella* spp.)”

The sensitivities in urine are 77 % (*E. coli*), 64 % (*Enterococcus* spp.), 50 % (*Klebsiella*). To avoid confusion with the sensitivity of the diagnostic test, we would prefer to not include these values in the manuscript. Still, the sensitivity was considered during the determination of optimal gating conditions as shown in Fig. S5c.

15. In companion diagnostic assay, the effects of bacterial burden in patient urine on the prediction (precision and sensitivity) should be discussed.

This mixed calculation is reflected by the F1-score, which is the harmonic mean of precision and sensitivity. All three values (F1-score, precision, and sensitivity) were taken into account when we determined the gating parameters (Figure S5).

Yes, there could be false-negatives due to low burden (discussed in lines 309-312) directly affecting sensitivity and to a lesser extent the precision. For a companion diagnostic test that should guide treatment decisions, the positive predictive value is most relevant.

This means that although a couple of potential responder patients could be missed due to lower sensitivity, most patients that are identified as responders will have a higher chance of treatment success.

The positive predictive values under standardized conditions are shown in Figure S5. Under the stringent killing activity cut off-of 0.2 (AUC ratio) they are the following:

78 % (*E. coli*), 89 % (*Enterococcus*) and 61 % (*Klebsiella*)

In urine (Figure 6), using the same parameters they change to:

78 % (*E. coli*), 93 % (*Enterococcus*) and 47 % (*Klebsiella*)

indicating that our gating conditions are quite robust and applicable for real-life samples with varying bacterial burden. Our dataset for *Klebsiella* (n=22) is smaller than for *E. coli* (n=45) and *Enterococcus* (n=36) which could partly explain the deficiency in robustness.

To avoid confusion of the readers we would like to only present the positive predictive value in patient urine (most relevant) and not provide a full comparison of patient urine vs. SHU beyond what is shown in Figure S5.

16. As a potential UTI detection technology, it would be useful to discuss the cost of phage-based detection compared with conventional methods. Such as the yield of viable phage (E2/E4/K1/K4/EfS3/EfS7) from bacterial culture, phage amounts for each sample detection, and luminescence materials/machine, etc.

While we agree that this is an interesting question, we provide a proof-of-concept and are not developing a commercial product. We therefore believe that commercial calculations are beyond the scope of the current study. However, non-pharmaceutical phage-based products are commercially viable, for example in the food industry, where margins are notoriously low (link).

Minor:

1. Why there are two no-phage controls with different cell-lysis dynamics in figure 2b (middle)?

We assume the reviewer is referring to Figure 3b. There are two no-phage controls, because these two phages were tested on their respective propagation strains (JH2-2 for EfS3 and EfS7 for EFS7). For clarification, we have now indicated the host strain in the revised Figure 3a-b.

2. Line 126, “the *nLuc* sequence was integrated alongside a ribosomal binding site (RBS) downstream of a strongly expressed endogenous promoter of phage structural genes.”

Figure 2 legend, “For *E. coli* and *Enterococcus* phages, codon optimized *nLuc* (516 bp + RBS) was incorporated immediately downstream of the major capsid gene (*cps*)”

Which one is correct? downstream of *cps* promoter or *cps* gene?

Thank you for this question. In fact, both is correct. For all phages, integration was downstream of a strong structural promoter, more specifically, it was downstream of *cps* for *Enterococcus* and *E. coli* phages and downstream of the prohead assembly protein gene for *Klebsiella* phages. All of these genes are controlled by strong promoters in the structural gene cassette. Figure 2a shows the precise integration sites for all phages. To clarify, we have changed the text to:

“The *nLuc* sequence was integrated alongside a ribosomal binding site (RBS) downstream of phage structural genes (Fig. 2a). Because the structural gene cassette is strongly expressed from endogenous phage promoters, this strategy confers a high level of reporter gene expression.”

3. Figures 3c and 4d, it would be better to indicate which controls are used in FC-RLU fold change calculation.

Good suggestion. We added the following sentence to both legends:

“Fold-change (FC) RLU equals RLU (sample) divided by RLU (phage only control).”

4. Line 255, “6 false positives (3 %) were observed across 206 samples and considering all three target pathogens”. Only 5 false positives were labeled in Figure 5b.

Thank you for spotting this error in the text. We corrected it in the text to:

“5 false positives (2.4 %) were observed across 206 samples and considering all three target pathogens”

5. Figures 6 and S5, for better understanding, i) Show the full names of TP/TN/FP/FN in S5 legend. “true positives (TP), true negatives (TN), false positives (FP) and false negatives (FN)”; ii) Show TP/TN/FP/FN information in Fig 6 b-d and legend. iii) Show the definition of Sensitivity/ Precision/ F1-score in the S5 legend. “Sensitivity ($TP/(TP+FN)$), Precision ($TP/(TP+FP)$) and F1-score ($2*TP/(2*TP+FP+FN)$) of the assay were calculated to define the RLU/AUC thresholds that yield the best prediction for antibacterial efficacy.” iv) What’s the significance of F1-score? Any reference?

Thank you for this suggestion, the following changes were implemented:

- i) added TP/TN/FP/FN to S5 legend.
- ii) added TP/TN/FP/FN to Fig. 6b-d and legend
- iii) added the definitions of sensitivity, precision, and F1-score to legend of Fig. S5.
- iv) added an explanation of F1 score to the main text: “The F1-score is the harmonic mean of precision and sensitivity and can be used to optimize trade-offs between these two metrics.”

Comments from Reviewer #3:

Review Report

Title of Manuscript:

Engineered Reporter phages for rapid detection of Escherichia coli, Klebsiella spp., and Enterococcus spp. in urine.

Overview:

The manuscript is interesting to read; it is original and showcases the innovation of a potential promising urine diagnostic assay for urinary tract infection (UTI) if developed further. Meile and co-workers demonstrated how they designed and evaluated a phage engineered luciferase dependent reporter system that detects three highly occurring uropathogen obtained from urine samples. Through genetic recombination, reporter-phage were designed with the intent to achieve a wide host coverage and to also induce bioluminescence in the presence of Escherichia coli, Klebsiella spp. and Enterococcus spp. Luminescence levels were quantified and used to determine the presence of uropathogens within 5 hours at a minimum threshold of $\geq 10^3$ CFU/ml. Analytical performance of the assay was assessed using infection kinetics, detection limit, detection range and potential cross reactivity. Clinical diagnostic performance characteristics such as specificities and sensitivities as well as overall accuracy were estimated. Assay was further evaluated for consideration as a companion test. The manuscript is publishable upon making corrections to important diagnostic parameters indicated in the comments made in this review.

We would like to thank the reviewer for the positive assessment of our work. Most of the differences in calculated performance values are due to an unclear description from our side, which we have clarified now (see below).

Title:

Title of manuscript identifies the study as an experimental one; this is appropriate and accurately conveys the principal information in the content of the paper.

Abstract

The abstract is not structured into subheadings but conveys the salient information required in an abstract. Parameters such as assay sensitivities, specificities and overall accuracies need to be re-

calculated and corrections effected in the abstract. Threshold levels for each assay must also be stated.

Introduction

Background, rationale, aim and objectives for the study were clearly communicated and appropriate references cited for literatures that were used.

Methods & Results

Appropriate techniques and procedures were adopted in the development of a recombinant reporter-phage with broad host range and fitted with a luciferase gene. Urine culture supplemented with Matrix-assisted laser desorption/ionization-time of flight mass spectrometry (MALDI-ToF MS) was used as gold standard assay for the estimation of performance characteristics of the newly developed urine phage-reporter assay using fresh urine sample collections from patients with urinary tract infection (UTI) cases and asymptomatic UTI. Methods adopted in analysing the results were valid and explanation given for their usage, however it is suggested that authors need to take note of the following and make the needed changes where necessary since this affects the interpretation, discussion and conclusions that may be drawn from the results:

Line 245 to 262 (Results section): Meile and Co-workers demonstrated that phage-reporter E2::nluc and E4::nluc was able to detect E. coli 40/59 (68%) while phage-reporters K1::nluc and K4::nluc was able to detect 22/27 (85%). The combined E. coli detections of 22 as reported here in the manuscript for the Klebsiella based phage-reporter assay should be changed to 23 detections (23/27). The following statement and calculation, provides further facts to suggest so, except otherwise and authors must clarify ...“Interestingly, while all 15 Klebsiella-containing samples from Site 1 were detected, the assay failed to identify 4 out of 12 positive samples from Site 2..... Furthermore, when you calculate $[(23/27)*100]$ it gives you the 85%.

Line 263 to 266 (Results section): Assay characteristics for the combined phage assays as reported in the manuscript for (E2::nluc and E4::nluc), (Efs3::nluc and Efs7::nluc) and (K1::nluc and K4::nluc) were respectively as follows for 206 urine samples collected from patients with urinary tract infection (UTI) cases and asymptomatic UTI:

- Sensitivity $[TP/(TP+FN) * 100]$: 68%, 85% and 78%
- Specificity $[TN/(TN+FP) * 100]$: 99%, 99% and 99%
- Accuracy $[TP/(TP+FP+TN+FN) * 100]$: 90%, 94% and 96%

The following calculated values for the performance characteristics of all the urine assays is misleading and need to be re-calculated again using the urine culture-based identification coupled to MALDI-ToF-MS (UCB-MALDI-ToF-MS) method as the gold standard method.

I illustrate below the calculations of the above parameters for the combined E. coli phage-reporter assay base on the results provided by the authors:

Please find our response below.

Test Result UCB-MALDI-ToF-MS

Positive Negative Total

E. coli urine-phage reporter Assay Positive TP (37) FP (3) TP+FP (40)

Negative FN (22) TN (144) FN+TN (166)

Total TP+FN (59) FP+TN (147) TP+FN+FP+TN (206)

Now, True Positive (TP) refers to the number of cases that tested positive for both the gold standard test and the new test. In the case of the *E. coli* assay, the results suggest that 40 cases were detected by the new assay out of the 59 cases detected by the gold standard test. However since 3 [False positive (FP)] of the 40 cases tested positive for the new test but negative for the gold standard test, the number of TP becomes 37.

Meaning of the following abbreviations as used above: True Negative (TN) and False Negative (FN).
From the table:

- Sensitivity $[(TP / (TP+FN)) * 100]$ for the combined *E. coli* assay (E2::*nLuc* and E4::*nLuc*) will now be 63% and this should replace the sensitivity of 68% recorded in the manuscript.
- Specificity $[(TN / (TN+FP)) * 100]$ for the combined *E. coli* assay (E2::*nLuc* and E4::*nLuc*) will now be 98 % and this should replace the specificity of 99 % recorded in the manuscript.
- Accuracy $[(TP+TN) / (TP+FP+TN+FN)) * 100]$ for the combined *E. coli* assay (E2::*nLuc* and E4::*nLuc*) will now be 88 % and this should replace the accuracy of 90 % recorded in the manuscript.

Performance characteristics for *Klebsiella* spp and *Enterococcus* spp. urine reporter assays are also supposed to be re-calculated as described above for the *E. coli*.

Furthermore, diagnostic performance characteristics in Figure 5c will need to be re-computed again.

We would like to thank the reviewer for a careful review of our data and for pointing out some unclarities in our descriptions. We agree that it was not evident from the Figure and text how the performance metrics were obtained. To clarify, we have now included all TP, FP, TN, and FN values in the revised Figure 5c (see below). In the process, we have found a small mistake in the calculations for the performance of the *Klebsiella* assay, which has been corrected during revision. Consequently, the performance metrics for *Klebsiella* have changed slightly (sensitivity: 85% → 87%; accuracy: 96% → 98%). All other metrics remained the same.

We have also improved the clarity of the text as it was not evident whether we were talking about TP or TP+FP values in some cases, which may have led to confusion. For example, we mentioned “results from individual E2::*nLuc* and E4::*nLuc* assays identified *E. coli* in 40 specimens (68 %)”. Now these 40 specimens were true positives and not TP+FP, which explains the different outcomes calculated for *E. coli* by the reviewer. To avoid confusion, we changed the text to:

“results from individual E2::*nLuc* and E4::*nLuc* assays led to 40 true positive identifications of *E. coli* (68 %).”

Below, please find our calculations for the performance metrics of all three reporter phage assays:

	E. coli			Enterococcus spp.			Klebsiella spp.		
	Site 1 	Site 2 	Both 	Site 1 	Site 2 	Both 	Site 1 	Site 2 	Both Total Nr. samples	147	59	206	147	59	206	147	59	206
True negative	103	42	145	123	36	159	128	47	175
True positive	27	13	40	18	17	35	18	8	26
False negative	15	4	19	4	6	10	0	4	4
False positive	2	0	2	2	0	2	1	0	1
Sensitivity	64%	76%	68%	82%	74%	78%	100%	67%	87%
Specificity	98%	100%	99%	98%	100%	99%	99%	100%	99%
Accuracy	88%	93%	90%	96%	90%	94%	99%	93%	98%

Figure 5c. Summary of the test performance showing calculated sensitivity, specificity, accuracy for all target pathogens at both sampling sites.

E.coli values

TP (40), FP (2), FN (19), TN (145)

Sensitivity (TP/(TP+FN)) = (40 / (40+19)) = 68 %

Specificity (TN/(TN+FP)) = (145/(145+2)) = 99 %

Accuracy (TP+TN) / (TP+FP+TN+FN) = (40 + 145)/(206) = 90 %

Enterococcus values

TP (35), FP (2), FN (10), TN (159)

Sensitivity (TP/(TP+FN)) = (35 / (35+10)) = 78 %

Specificity (TN/(TN+FP)) = (159/(159+2)) = 99 %

Accuracy (TP+TN) / (TP+FP+TN+FN) = (35 + 159)/(206) = 94 %

Klebsiella values

TP (26), FP (1), FN (4), TN (175)

Sensitivity (TP/(TP+FN)) = (26 / (26+4)) = 87 %

Specificity (TN/(TN+FP)) = (175/(175+1)) = 99 %

Accuracy (TP+TN) / (TP+FP+TN+FN) = (26 + 175)/(206) = 98 %

Discussion & Conclusion

Discussion is clear however authors are to tone down a bit on findings related to the diagnostic performances.

We have adapted the performance metrics throughout the manuscript.

REVIEWERS' COMMENTS

Reviewer #2 (Remarks to the Author):

The authors have thoroughly addressed my comments and made the necessary modifications to the manuscript. I am satisfied with their response and believe the paper has been significantly improved. I am pleased to recommend the manuscript for publication.